# Transposon signatures of allopolyploid genome evolution

Adam M. Session [1,2,3] ✉ & Daniel S. Rokhsar [1,2,4,5]

Hybridization brings together chromosome sets from two or more distinct progenitor species. Genome duplication associated with hybridization, or allopolyploidy, allows these chromosome sets to persist as distinct subgenomes during subsequent meioses. Here, we present a general method for identifying the subgenomes of a polyploid based on shared ancestry as revealed by the genomic distribution of repetitive elements that were active in the progenitors. This subgenome-enriched transposable element signal is intrinsic to the polyploid, allowing broader applicability than other approaches that depend on the availability of sequenced diploid relatives. We develop the statistical basis of the method, demonstrate its applicability in the well-studied cases of tobacco, cotton, and *Brassica napus*, and apply it to several cases: allotetraploid cyprinids, allohexaploid false flax, and allooctoploid strawberry. These analyses provide insight into the origins of these polyploids, revise the subgenome identities of strawberry, and provide perspective on subgenome dominance in higher polyploids.

Polyploidy is common in plants and some animal groups[1]; indeed, all angiosperms and vertebrates are descended from polyploid ancestors[2–4]. Broadly speaking, there are two kinds of polyploids: those that form by genome doubling within a species (autopolyploidy) and those that form by genome doubling in association with interspecific hybridization (allopolyploidy)[5–7]. In autopolyploids, each chromosome can choose among multiple meiotic partners allowing recombination among equivalent homologous chromosomes and producing polysomic inheritance (more than two alleles per locus); diploidy and disomic inheritance may be restored by the subsequent evolution of pairing preferences[8,9]. In allopolyploids, however, genome doubling associated with interspecific hybridization ensures that all chromosomes have defined homologous meiotic partners derived from their respective progenitors. This feature allows the parental chromosome sets to be stably maintained by disomic segregation without recombination between homoeologous chromosomes. In contrast, in homoploid hybrids (i.e., interspecific hybridization without genome doubling) recombination between homoeologous chromosomes shuffles the genetic contributions of the progenitor species, and

subgenomes generally cannot persist as stable entities except in rare cases of asexual reproduction[10] or fixed translocation heterozygosity preventing the production of viable recombinants[11]. Thus, we can define homoeologs as chromosomes that have diverged by evolution in different species but are ultimately derived from the same ancestral chromosome (Fig. 1a).

The stable chromosome sets that descend from distinct ancestral progenitors are referred to as the subgenomes of an allopolyploid[1,12]. A recurring challenge in the analysis of allopolyploid genomes is recognizing the chromosomes that belong to each subgenome, and, when possible, identifying its specific diploid progenitor. The conventional method for proving allopolyploidy and characterizing the resulting subgenomes relies on establishing phylogenetic relationships between the protein-coding genes of a polyploid and extant diploid relatives[13,14]. In allotetraploids (with two subgenomes), it is sufficient to find an extant relative of one diploid progenitor, since the other subgenome can be inferred by the process of elimination even in the absence of a corresponding diploid relative[13,15] (Fig. 1b).

[1]Department of Molecular and Cell, University of California, Berkeley, CA 94720, USA. [2]US Department of Energy Joint Genome Institute, 1 Cyclotron Road, Berkeley, CA 94720, USA. [3]Department of Biological Sciences, Binghamton University, Binghamton, NY 13902, USA. [4]Molecular Genetics Unit, Okinawa Institute for Science and Technology Graduate University, Okinawa, Japan. [5]Chan Zuckerberg BioHub, San Francisco, CA, USA. ✉e-mail: asession@binghamton.edu

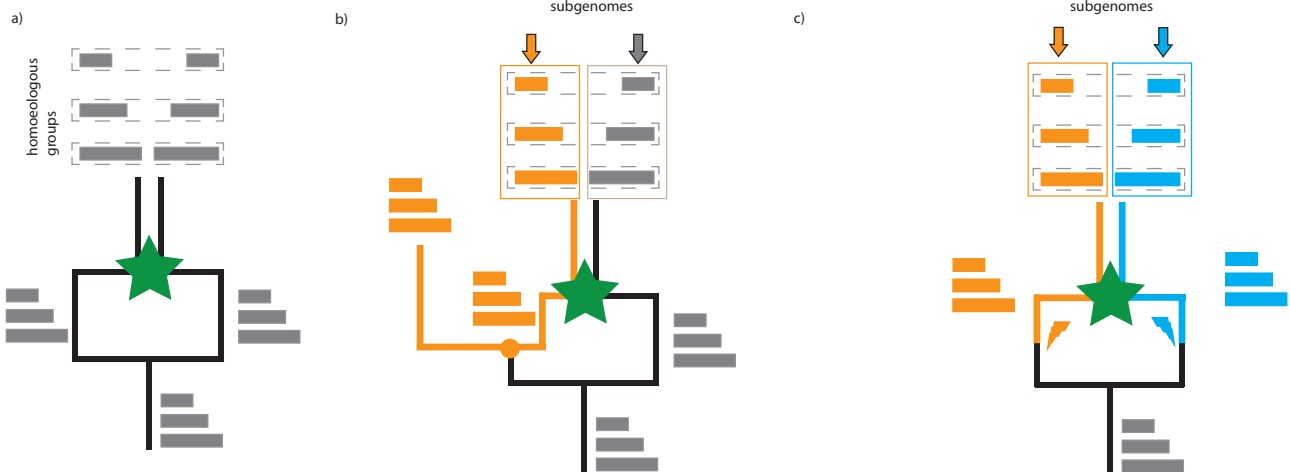

**Fig. 1 | Allotetraploids and subgenomes.** In each panel, time is increasing in the vertical direction. At the bottom, the monoploid chromosome set of the original diploid ancestor is shown as three horizontal gray bars of different lengths (i.e., $x = 3$), and green stars represent subsequent allotetraploidization events (inter-specific hybridization combined with genome doubling). **a** Hybridization brings together homoeologous chromosome sets. In the present day allotetraploid (top), homoeologous chromosome pairs (horizontal dashed rectangles) can be recognized by their sequence similarity, but their subgenome identity (i.e., whether they are derived from the left or right progenitor) cannot be determined without further information. This lack of subgenome information is indicated in gray. **b** if a diploid relative of one of the progenitors is known, chromosomes descended from this progenitor can be recognized in the tetraploid (here chromosomes colored orange have demonstrated shared ancestry with a diploid relative (orange branch). In an allotetraploid, the remaining chromosomes shown in gray define the second sub-genome, by exclusion. Chromosomes grouped into subgenomes are surrounded by vertical orange and gray boxes. Speciation between the diploid relative and one of the progenitors of the tetraploid is shown as an orange circle. **c** Independently evolving progenitors (left and right branches) are expected to accumulate unique transposable element activity (shown as orange and blue lightning bolts) that mark their respective chromosome sets. In the resulting allotetraploid, the asymmetric distribution of transposon-derived repetitive elements can be used to partition chromosome sets into subgenomes (vertical orange and blue boxes) that contain one member of each homoeologous pair. In this analysis no external diploid comparison is needed, since the sub-genome signal is intrinsic to the tetraploid genome sequence itself.

There is no guarantee, however, that diploid progenitor lineages of ancient polyploids still exist; indeed, polyploids may outcompete their diploid relatives and contribute to their extinction[16]. This appears to be the case, for example, for the paleo-allotetraploid frog *Xenopus laevis*[17] and the giant grass *Miscanthus* spp[18]. Even when related diploid lineages do exist, it may be difficult to definitively identify them or relate them to a specific subgenome, due in part to the challenge of phylogenomically resolving a rapid radiation of potential progenitors. Cultivated false flax, an allohexaploid, required extensive sampling of diverse populations in order to correctly resolve the evolutionary history of its chromosomes via traditional methods[19,20]. Cultivated strawberry, which is an allooctoploid, has proven to be particularly difficult to resolve, in part due to the rapid radiation of diploid strawberries[21–23]. Finally, subgenomes may become rearranged during polyploid evolution, and scenarios must account for some degree of homoelogous exchange or replacement between subgenomes, as has been documented in numerous polyploids[18,24–28].

Other features intrinsic to a polyploid genome can be used to develop hypotheses about allopolyploidy and subgenome identity when suitable extant diploid genomes are not available. The most common approach takes advantage of the phenomenon of biased fractionation, that is, preferential loss of homoeologous genes on one subgenome relative to the other due to asymmetric gene silencing or deletion[29–31]. Biased gene loss between homoeologous chromosomes is generally taken as *prima facie* evidence for allopolyploidy, and chromosomes with high and low gene retention rates can be plausibly assigned to distinct subgenomes[29]. While suggestive, such inferences do not definitively prove allopolyploidy since in theory other asymmetric processes could lead to differentiation between pairs of homoeologous chromosomes without systematic biases based on species of origin. Furthermore, some allopolyploids show no significantly biased gene loss, perhaps due to recent formation or close similarity between progenitors[32–36]; in such cases biased fractionation cannot be used to infer subgenomes.

Another relevant signal explored below is based on the observation is that even closely related species are often marked by the unique and characteristic activity of specific transposable elements (TEs), which expand and distribute copies across nuclear genomes in irregularly timed bursts of activity[37–45]. Copies of transposable elements inserted into chromosomes become durable markers of recent evolutionary history that persist after allopolyploidy (Fig. 1). The chromosomal distribution and timing of transposon insertions therefore provides a tracer of chromosome history in polyploids[17,18,36,46,47]. In particular, chromosomes inherited from the same progenitor, i.e., subgenomes, are expected to share a common set of repetitive elements not found on chromosomes inherited from different progenitors.

Importantly, this history of transposon activity is recorded in the polyploid genome itself (Fig. 1c), and can be recovered without comparison to extant diploids[17]. This feature is particularly useful when the relevant diploid lineages are extinct, unsampled, not sequenced, or poorly phylogenetically resolved due, for example, to incomplete lineage sorting. When available, relevant diploids can offer additional support for subgenome-specific transposon expansions (as we discuss below for allooctoploid strawberry). Subgenome-specific TEs serve as more robust markers of chromosome ancestry than biased fractionation since they are established in the progenitors themselves and do not depend on subsequent post-polyploidy changes. In higher polyploids, TE expansions in intermediate polyploid progenitors also enable us to infer the order of hybridization. Furthermore, as shown below subgenome-specific markers derived from TE activity allow us to identify post-hybridization translocations between subgenomes even in genomic regions where protein-coding genes are sparse[18].

Here, we develop a statistical framework for identifying evolutionarily coherent subgenomes within allopolyploid genomes that relies on transposable elements to group chromosomes into sets with shared ancestry, without regard to any external genome comparisons. This approach has been initially applied in an ad hoc fashion in several

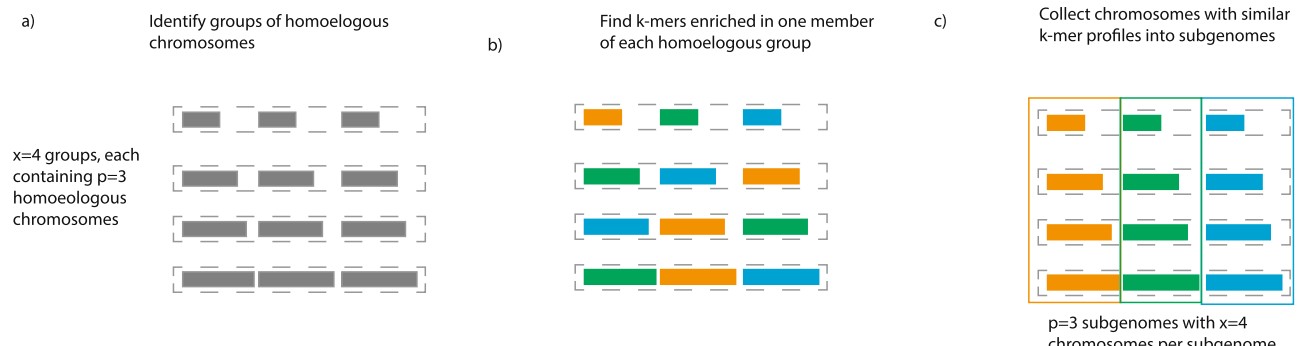

**Fig. 2 | Outline of method.** Schematic of our approach as applied to a hexaploid ($p$ = 3 chromosome sets) with $x$ = 4 chromosomes per set. As in Fig. 1, chromosomes are colored according to their inferred subgenome identity. **a** First, we identify groups of homoeologous chromosomes based on shared protein-coding genes (horizontal dotted boxes). At this stage, the subgenome identity of each chromosome within a homeologous is not known (indicated by gray color). **b** Next, we identify nominally subgenome-specific k-mers that are enriched in one member of each homoeologous group. For a hexaploid, there are three different identities represented by orange, green, and blue corresponding to three sets of differentially enriched k-mers. Each homoeologous group has one chromosome of each color. **c** Finally, we collect chromosomes with similar k-mer profiles into subgenomes (shown as chromosome sets in three vertical boxes). The k-mers selected in (**b**) are markers for transposable elements with distinct evolutionary histories (lightning bolts in Fig. 1c).

allotetraploid genomes[17,18,36,46,48–50]. We provide a statistical basis for the method and extend its applicability to higher polyploids. After outlining the general method, we first demonstrate its utility by applying it to multiple well-known allotetraploids before considering two higher ploidy cases. For allohexaploid false flax, *Camelina sativa*[19,20,51,52], we identify sets of repeats that serve as a positive marker of diploid and allotetraploid ancestors of *C. sativa*. Finally, we turn to allooctoploid strawberry, *Fragaria* x *ananassa*, whose proposed subgenome assignments have been disputed based on genome comparisons with related diploids[21–23,53–56]. We confirm that two of the four octoploid strawberry subgenomes are derived from the diploid lineages of *F. vesca* and *F. iinumae*[21,53,55–58], but we find a partitioning of the remaining fourteen chromosomes into two subgenomes that differs from previous hypotheses[58,59].

## Results and discussion
### Overview of methodology
Our approach to recognizing the chromosomes belonging to an evolutionarily coherent subgenome based on the distribution of repetitive elements was inspired by inference of authorship for unsigned essays in The Federalist Papers[60]. In an early application of Bayesian statistics, Mosteller and Wallace quantified differences in word usage among essays attributed to Federalist authors Alexander Hamilton, James Madison, and John Jay. By combining these individually weak word-by-word signals, Mosteller and Wallace robustly identified the author of anonymously published essays. By analogy, in the case of polyploid genomes each subgenome is also written by a different author (i.e., progenitor), and distinctive DNA word usage between subgenomes is due to past transposon activity. Unlike the authorship problem, however, in the subgenome identification problem we may not have a training set of chromosomes of known diploid provenance. We show below how to bootstrap the identification of discriminatory DNA words from chromosome comparisons even in the absence of a training set.

By analogy with the authorship problem, we seek short DNA words of a defined length k (k-mers) that serve as markers for subgenome-enriched families of repetitive elements[18,36,46,48–50]. As a practical matter, we typically use $k$ = 13. These repetitive words are intrinsic features of the polyploid genome sequence, and our method does not depend on information from lower ploidy relatives, although such information can also be integrated (see the discussion of strawberry below). The use of repetitive sequences to identify subgenomes in polyploids is an increasingly used methodology[17,18,36,46,50,61,62] that is complementary to, and can resolve errors arising from, protein-coding

analyses that rely on external progenitor surrogates. Notably, our approach enables statistical testing of alternate subgenome hypotheses based on asymmetric distribution of repetitive elements, e.g., using Tukey's range test[63].

### General partitioning of chromosomes into subgenomes
We consider an allopolyploid with $2p$ chromosome sets per somatic cell and $x$ chromosomes per set, so that $2n = 2px$. The gametic complement of $n = px$ chromosomes can generally be divided into $x$ homoeologous groups with $p$ chromosomes per group, where $x$ is typically the basic chromosome number of the progenitors (Fig. 2a). (This formula is easily modified in the presence of additional chromosomal rearrangements, as discussed for *Camelina* below.) For allotetraploids the homoeologous groups are pairs ($p$ = 2), for allohexaploids they are triplets ($p$ = 3), and for allo-octoploids they are quartets ($p$ = 4). Homoeologous chromosomes are easily recognized by their enrichment in paralogous genes, often with substantial collinearity for recently formed polyploids. Chromosomal homology to distantly related diploids (i.e., outgroups to the polyploidization process) may also be useful in identifying homoeologs. The problem of assigning chromosomes to subgenomes amounts to simultaneously labeling the members of each homoeologous group so that one member is assigned to subgenome 1, another is assigned to subgenome 2, etc. Since the label given to each subgenome is arbitrary, there are $(p!)^{x-1}$ distinct possible partitions of a polyploid genome. Rearrangements relative to the progenitors may obscure this organization but are readily accommodated by our method (see, e.g., the case of *C. sativa* and *Nicotiana tabacum* below).

To identify subgenomes we therefore seek to partition chromosomes into non-overlapping sets based on repetitive signals of shared ancestry (Fig. 2b, c). Given the dynamic nature of transposable elements even within closely related populations or species[36,39,40,64], TEs may represent the majority of novel DNA in closely related species, and we expect that the chromosomes belonging to each subgenome will share repetitive content inherited from its progenitor. Rather than try to identify and classify such subgenome-specific transposable element families directly, we use commonly occurring k-mers as markers of repetitive sequence[17,18,36,46,48–50]. In general, we have found that $k$ = 13 provides a balance between k-mers that are too short (in which case many high copy k-mers will either be common by chance, or due to overlapping with microsatellite expansions that are not useful markers for subgenome-specific activity) or too long (since longer stretches of DNA are more likely to be disrupted by a mutation after polyploid formation, making them less useful as markers for ancient TEs

activity). Given an annotation of transposable element content of a genome, it is a simple matter to identify those TE families that overlap subgenome-specific k-mers.

Operationally we begin with chromosomes (or chromosome segments) grouped into homoeologous pairs, triples, quartets, etc. as appropriate for tetraploid, hexaploid, octoploid, etc. genomes. We then (1) identify k-mers that have high copy number across the entire polyploid genome (i.e., k-mers with more than $N_{min}$ copies), (2) scan each homoeologous group to identify high copy k-mers that are enriched in one homoeolog vs. at least one of the other members of that group (by a factor F), and (3) select for further consideration those high-copy k-mers that show such enrichment across multiple homoeologous groups. Condition (1) ensures that the k-mers mark repetitive elements; condition (2) identifies k-mers whose distributions are asymmetric across homoeologs, and so represent a potential subgenome marker; and condition (3) focuses attention on k-mers that are consistently distributed across multiple non-homoeologous chromosomes, which can unite such chromosomes into a subgenome. (It is often the case, especially in draft genomes, that individual chromosomes have specific pericentromeric k-mer expansions not found in other chromosomes, but these are not useful for our purposes.) We emphasize that the discovery of high copy k-mers that are potentially subgenome-enriched depends only on intragenomic comparisons among homoeologous chromosomes and does not depend on any prior knowledge derived from external datasets or hypotheses about how these homoeologs are grouped into subgenomes.

We organize chromosomes into subgenomes by hierarchical clustering based on their shared content of potentially subgenome-enriched high copy k-mers, using 1-r as the distance measure, where r is Pearson's correlation coefficient. Each of the p clusters correspond to chromosome sets of shared ancestry. If a progenitor experienced a species-specific expansion of repetitive elements, then we expect to find a corresponding set of k-mer markers enriched in the subgenome descended from that progenitor, relative to one or more of the others. In the case of rearranged genomes, this analysis can be performed on homoeologous chromosome segments rather than entire chromosomes[18]. If desired, other related genomes can also be included, since the subgenome-enriched k-mers may also be present in other species that share the same transposon activity. We emphasize that the hierarchical clusterings of chromosomes shown in heatmaps in Figs. 2–5 are not phylogenetic relationships, but rather groupings based on shared repetitive content. In cases of higher ploidy, we might expect to find k-mer markers that are associated with two or more subgenomes. These possibilities are explored further below in our case studies.

Given the groupings of chromosomes into subgenomes that are identified by this clustering process, we can refine our collection of subgenome-enriched k-mer markers using ANOVA and Tukey's range test (shown as volcano plots) in a manner analogous to analyses of differential gene expression between conditions[63,65]. Our data satisfies all the assumptions of ANOVA[66]: (1) the log-transformed k-mer count/bp is normally distributed across each subgenome discussed in this paper (Supplementary Fig. 1); (2) the distributions within each species have roughly equal variance; and (3) the measurements of k-mer count in each chromosome are independent from one another. In this way we can identify subgenome-enriched k-mers beyond the heuristic fold-enrichment cutoffs, supported by Bonferroni-corrected p-values (Supplementary Note 1–7). For strawberry, Camelina, and tobacco we explicitly relate the subgenome-specific k-mers to corresponding LTR retrotransposons (Supplementary Data 1–5, For strawberry, we use the divergence between 5'- and 3'-LTRs to infer the timing of this retrotransposon activity and the order in which the subgenomes were added to form the ultimate octoploid genome.

After, or even during, allopolyploid formation, homoeologous chromosome segments may become exchanged or even replace each

other[67], and other inter-chromosomal rearrangements may also occur[31,67]. If this process is extensive, it may erase the subgenome structure of the polyploid. Limited rearrangement, including homoeologous exchange and/or replacement, however, can be detected by our method as discrete segmental variation in subgenome-specific k-mer content along a chromosome. We implement this approach by using a Hidden Markov Model (HMM) to predict the subgenome identity of chromosomal segments based on the density of the subgenome-enriched k-mers (Supplementary Note 8). Inter-chromosomal rearrangements are discussed below for allotetraploid tobacco, and our HMM results agree with findings based on comparisons with diploid genomes and ad hoc k-mer based approaches applied to fragmented assemblies[31]. The other polyploids discussed in this paper, including strawberry, did not exhibit large-scale rearrangements between subgenomes detected by this method.

A computational toolkit for applying these methods, and a worked example using tetraploid *Brassica napus* can be found at Github (https://github.com/amsession/Kmer-based-Subgenome-Mapping)[68] (Supplementary Note 1, Supplementary Fig. 2). Raw data and statistics for all 13-mers for *B. napus* analysis are provided in Supplementary Data 4 and 6.

## Shared allotetraploidy in cottons

We first demonstrate the utility of our method using several well-studied allotetraploid plant and animal genomes. Cotton (*Gossypium*) is one of the best studied allopolyploid model systems available, with multiple allotetraploid species ($n = 26$, $x = 13$, $p = 2$) that have previously been shown to descend from a common allotetraploidy event between two diploid progenitor species[28,33,69–71]. In particular, *Gossypium hirsutum* (AD1) and *G. barbadense* (AD2), each contain an A subgenome, related to the African diploid *G. arboretum*, and a D subgenome, related to *G. raimondii*[70]. The A and D progenitor species diverged roughly 5–10 million years ago (mya) while allotetraploid cotton emerged in the last 1–2 million years[71].

We clustered the chromosomes of the two tetraploid cottons based on k-mer counts, both individually and together, without using any information from diploid cottons (for each species, we used Tukey's HSD, df = 24, Bonferroni-corrected $p < 0.05$; Supplementary Note 2; Fig. 3, Supplementary Fig. 3). We find that (1) each tetraploid species naturally splits into subgenomes, and (2) that the subgenomes in the two tetraploids species are orthologous. We identified 320,481 13-mers enriched on the A subgenome relative to D of AD1, and 311,195 13-mers enriched on the A subgenome in AD2. Similarly, there are 94,305 13-mers enriched on the D subgenome of AD1, while 86,379 13-mers are enriched on the D subgenome of AD2. 276,264 A 13-mers and 71,988 D 13-mers are shared between both tetraploid *Gossypium* species. The cross-species association between subgenomes of AD1 and AD2 reflects the sharing of subgenome-enriched k-mers, which implies that they share the same (or closely related) progenitors, consistent with previous comparative analysis of the tetraploids with diploid *Gossypium*[70]. Our method bypasses the need for comparison with diploid species, however, allowing it to be applied in cases where the diploid progenitors are not known or the polyploid species has outcompeted its diploid ancestors. Raw data and statistics for all 13-mers for cotton analysis are provided in Supplementary Data 4, 7, and 8.

## Shared allotetraploidy in cyprinid fishes

Similarly, goldfish (*Carassius auratus*) and common carp (*Cyprinus carpio*) are related allotetraploid freshwater cyprinid fish with the same karyotype ($2n = 100$, $x = 25$, $p = 2$) that diverged ~11 mya[72]. Interest in their evolution and domestication has spurred the determination of chromosomal genome sequences for both common carp[15,73] and goldfish[61,72,74]. Previous phylogenetic analyses of protein-coding genes showed that diploid barbels ($2n = 50$) are more closely related to one

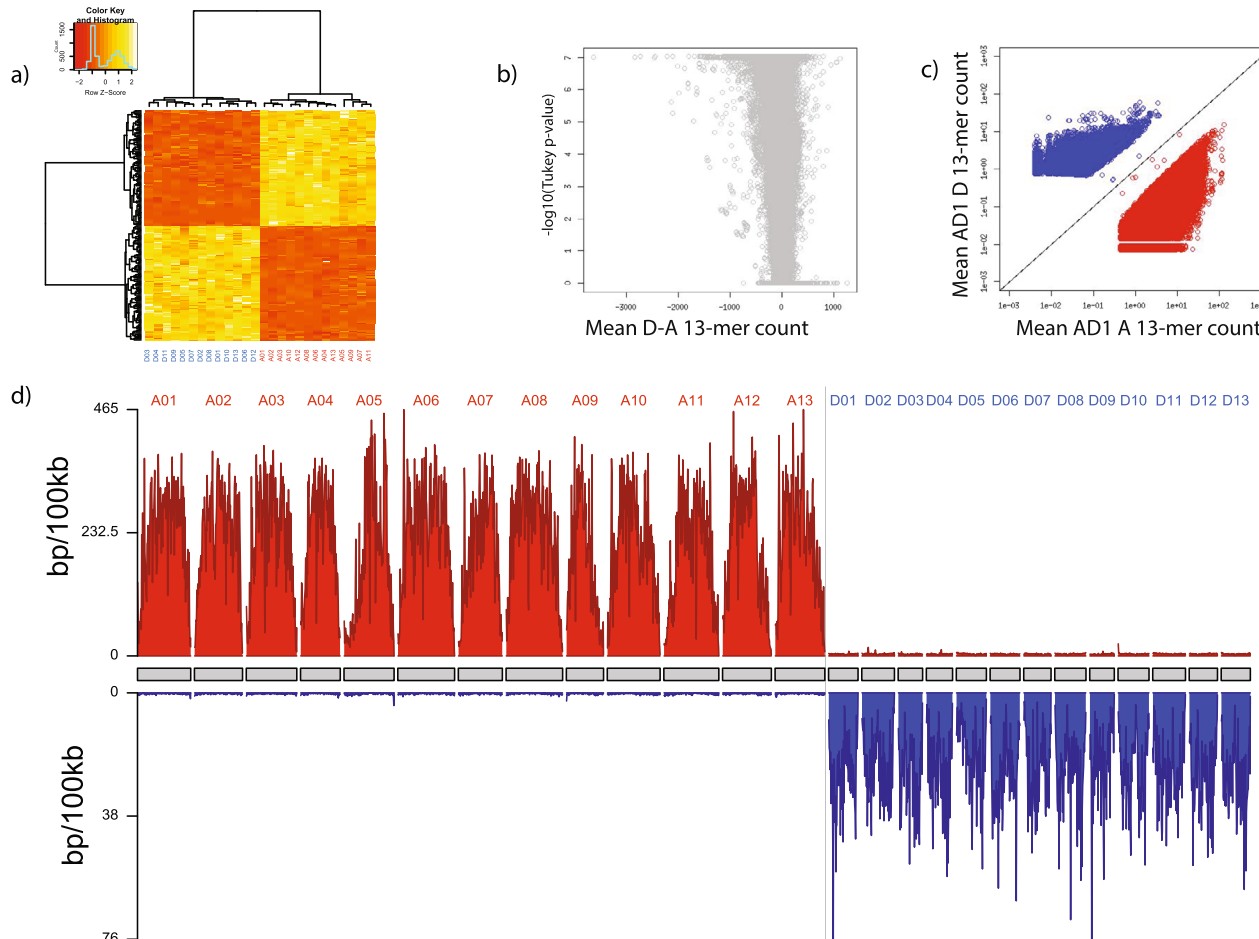

**Fig. 3 | Cotton allotetraploidy.** Upland cotton (*Gossypium hirsutum*) is an AADD allotetraploid, where the A subgenome related to African diploids and the D subgenome is related to *G. raimondii*[33]. It is called AD1 to differentiate it from the related tetraploid *G. barbadense* (AD2). We sought to identify k-mers that differentiate the subgenomes of *G. hirsutum* (Supplementary Note 2; for *G. barbadense* see Supplementary Fig. 3). **a** Heatmap showing 13-mer density as a function of AD1 chromosomes (columns, clustered on top) vs. a sample of 100 13-mers found to differentiate the A- and D-subgenomes (rows, clustered on left). A chromosomes are indicated in red and D chromosomes are indicated in blue. **b** Volcano plot showing Bonferroni-corrected Tukey *p*-value (Bonferroni-corrected; df = 24) vs. mean 13mer count difference D-A between subgenomes of *Gossypium hirsutum*. Each point is a 13mer. By definition, all Tukey's HSD tests are one-sided. Effect size is

shown on the *x*-axis (converted to 13mer count/chromosome) and 95% Confidence Intervals for each 13mer can be found in Supplementary Data 4. **c** Scatterplot showing mean A chromosome 13-mer count on *x*-axis, mean D chromosome 13-mer count on the *y*-axis. A-enriched 13-mers shown in red, D-enriched 13-mers shown in blue. Black line is *y* = *x*. Only the 13-mers found to differentiate subgenomes (Bonferroni-corrected *p* < 0.05) are shown. **d** Karyogram showing density of A-enriched (above, red) and D-enriched (below, blue) 13-mers in 100-kb bins along each of the chromosomes of *G. hirsutum*. The densities shown here only count 13-mers with at least a 100x bias to avoid showing the weakly enriched but statistically significant 13-mers that lie near the equal line in **c**. Source data are provided as a Source Data file.

chromosome set of common carp[15] and of goldfish[74] than the other, defining robust A-vs-B goldfish subgenomes and P-vs-M carp subgenomes. (Here A/B and P/M are the names given to the goldfish and carp subgenomes in respective original publications). This use of an extant diploid to identify one subgenome of a tetraploid is a practical example of the method diagrammed in Fig. 1b. For goldfish the barbel-derived subgenome M was inferred to be maternal (relative to the interspecific hybridization underlying allotetraploidy) based on mitochondrial DNA comparisons. Finally, the goldfish genome was also partitioned into L-vs-S subgenomes based on transposable elements[61] using the method of Fig. 1c[17]. Although early comparisons of draft genome sequences were equivocal[72], the chromosomes of common carp and goldfish are now understood to be in a 1:1 relationship[15,74] with subgenomes related as A = P = S and B = M = L, supporting a common allotetraploid ancestor as originally hypothesized[72,75–77].

We tested our method to see if it could recover these results from the allo-tetraploid carp and goldfish genomes alone, without reference to any diploid genomes. By clustering the chromosomes of goldfish

and common carp separately for each species (Supplementary Note 3; Supplementary Fig. 4 and 5), and together (Supplementary Fig. 5c) we find that (1) each genome naturally splits into subgenomes based on shared repetitive 13-mers, and (2) that the subgenomes in each species are associated with each other, as also observed for the better studied cotton duplication discussed above (Supplementary Fig. 5c). The association between subgenomes of common carp and goldfish reflects the sharing of subgenome-enriched k-mers between these species, which implies that they share the same (or closely related) progenitors, and likely arose from a common allotetraploidization event, similar to the case of cotton. These results are consistent with previous observations that carp and goldfish share at least one subgenome based on the phylogenetic analysis of genes discussed above[15,74].

We identified 185 A/P/S and 822 B/M/L candidate subgenome-specific markers, requiring a minimum mer-count of $N_{min} = 100$ and F = 2-fold enrichment for 24 out of the 25 homoeologous chromosome pairs (for each species, we used Tukey's HSD df = 46;

Bonferonni-corrected $p < 0.05$; Supplementary Fig. 4 and 5). We use a relaxed condition of 24 out of 25 chromosome pairs because goldfish chromosome GF33 is an outlier that is unexpectedly short and has few protein-coding genes or annotated repeats[61]. Its homoeolog, GF8, displays a positive B/M/L signal (Supplementary Fig. 5d). The orthologous chromosome pair in carp, CC16/CC15, display positive A/B signals, respectively. These results are consistent with GF33 being an A/P/S chromosome whose repetitive and protein-coding content may not be well-assembled[61]. Raw data and statistics for all 13-mers for cyprinid analysis are provided in Supplementary Data 4, 9, and 10.

## Allohexaploidy in the oilseed crop *Camelina sativa*

The oilseed crop *Camelina sativa*, also known as false flax, is an allohexaploid with disomic inheritance ($2n = 40$, $p = 3$, $x \sim 7$). (Here the chromosome number of the hexaploid is not a simple multiple of 7 since one chromosome arose by fusion of two ancestral *Camelina* chromosomes. Thus $n = 7 + 7 + 6 = 20$). Genome sequencing confirmed its triplicated genetic content compared with the related diploid *Arabidopsis lyrata*[19,20,78]. In the initial analysis of the *C. sativa* genome[19,51] the hexaploid chromosome set was partitioned into three putative subgenomes based on similarity to *A. lyrata*. More recently, however, comparisons of *C. sativa* chromosomes with sequences of a diverse group of diploid and tetraploid *Camelina* species were used to partition the hexaploid into subgenomes SG1, SG2, and SG3[20,52] that differ from those originally hypothesized from analysis of the hexaploid genome alone. In particular, these more recent studies established diploid *C. hispida* ($n = 7$, referred to as $H^7$) as one likely progenitor of *C. sativa*, and tetraploid *C. intermedia* (formerly *C. macrocarpa*) ($n = 13$) as the other[52].

Furthermore, the diploid *C. neglecta* ($n = 6$, referred to as $N^6$) was found to be an extant relatives of tetraploid *C. intermedia* ($n = 13$; $N^6N^7$), where an unknown or extinct diploid *C. hispida*-like ancestor with $n = 7$ chromosomes (referred to as $N^7$) was inferred. Thus the $n = 20$ hexaploid *C. sativa* can be represented as $N^6N^7H^7$, consistent with cytogenetic analysis[51,52], with SG1 and SG2 corresponding to the two N-type subgenomes and SG3 as the H-type subgenome.

We asked whether the ground truth established by the cytogenetic and comparative genomic analyses of Mandakova et al.[51,52] and Chaudhury et al.[20] could have been recognized directly from the hexaploid genome without reference to extant lower ploidy genomes. Due to the chromosomal rearrangements in *C. sativa* documented by Kagale et al.[19], we first identified candidate subgenome-enriched 13-mers by considering three clearly paralogous chromosome trios ((Csa15, Csa19, Csa01); (Csa17, Csa14, Csa3); and (Csa04, Csa06, Cs09)), without reference to other chromosomes or any comparative data. Candidate 13-mers from this three-trio analysis yielded a consistent partitioning of all chromosomes into three subgenomes (Supplementary Note 4). Finally, we refined the catalog of subgenome enriched 13-mers using ANOVA and Tukey's range test analysis (Tukey's HSD; df = 17; Bonferonni-corrected $p < 0.05$); Supplementary Note 4; Fig. 4; Supplementary Fig. 6h-i).

The three *C. sativa* subgenomes discovered by this approach are in direct correspondence with SG1 ($N^6$), SG2 ($H^7$), and SG3 ($H^7$) inferred by Chaudhary et al. comparative analysis[20], and consistent with Mandakova's et al. cytogenetic studies[51,52] (which did not differentiate SG1 and SG2). We find 2,783 13-mers that are systematically enriched on SG3 relative to SG1 and SG2, and conversely 714 13-mers that are enriched on SG1 and SG2 relative to SG3, but only 27 13-mers enriched on SG2 and 158 13-mers enriched on SG1 (Fig. 4a;Supplementary Note 4). In addition to identifying the three subgenomes using only the information from the hexaploid *C. sativa*, the shared repeat signal between SG1 and SG2 implies that these two subgenomes were contributed by an allotetraploid ancestor, consistent with, but logically independent of, comparisons with the related tetraploid *C. intermedia*[20,52](Fig. 4d). Raw data and statistics for all 13-mers for

*Camelina* analysis are provided in Supplementary Data 4 and 11. *Camelina* retrotransposons and list of families are in Supplementary Data 1, while alignments are included in Supplementary Data 5.

## Deciphering the ancestry of octoploid strawberry

Finally, we consider the controversial subgenome structure of the cultivated octoploid strawberry *Fragaria* x *ananassa* ($2n = 8x = 56$; $p = 4$, $x = 7$)[21-23,54-59,79], whose genome was recently sequenced[58]. Octoploid strawberry exhibits disomic inheritance (i.e., consistent pairing in meiosis) which suggests that four subgenomes descended from four different $x = 7$ progenitors likely persist. There is widespread consensus, based on comparisons of the protein-coding genes of octoploid strawberry with those of extant diploid relatives, that one of the four subgenomes, designated V, is closely related to the diploid woodland strawberry *F. vesca*[21,23,55-58], and a second subgenome, designated I, is closely related to *F. iinumae* from Japan and eastern Russia[80]. There is, however, no consensus definition of the other two subgenomes of octoploid strawberry or their corresponding diploid progenitors[21-23,53-58,79,81] (summarized in ref. 59.). Indeed, the preservation of well-defined subgenomes has been called into question based on claims of extensive homoeologous exchange based on analysis of protein-coding gene phylogenies[21-23,58]. Since our approach is agnostic to prior hypotheses about progenitors or subgenome identity, it is particularly well-suited for addressing these issues.

K-mer analysis reveals four clearly defined subgenomes in allo-octoploid strawberry (Fig. 5a;Supplementary Fig. 6-8; Supplementary Note 5-7). Each subgenome is a complete set of $x = 7$ chromosomes (i.e., one chromosome from each homoeologous quartet), that shares a characteristic pattern of past repetitive activity as recorded by a specific combination of enriched k-mers. Specifically, we found 829 subgenome-enriched 13-mers with $N_{min} = 100$ and F = 2 that form five classes (see side-clustering on Fig. 5a). Since there are more than two subgenomes, shared ancestry among subgenomes can arise during the progressive formation of octoploid strawberry from successive hybridizations of lower ploidy progenitors. Thus, some k-mer classes correspond to specific combinations of subgenomes. This finding parallels the discovery of k-mers enriched in SG1 and SG2 relative to SG3 in hexaploid *C. sativa*.

Two of the octoploid strawberry subgenomes defined by the k-mer enrichment method clearly correspond to the well-established V and I subgenomes, which are defined by specific 13-mer markers. The robustness of these subgenomes is further supported by a k-mer clustering of octoploid strawberry with the chromosome-scale genomes of the diploids *F. vesca*[82] and *F. iinumae*[22] (Fig. 5a). Although the 13-mers used in this clustering are defined only based on the octoploid genome, they also group the chromosomes of both diploid strawberries with their corresponding subgenomes of octoploid strawberry based on shared repetitive content. Again we emphasize that Fig. 5a shows hierarchical clusterings based on repetitive content and not a phylogenetic tree.

The other two chromosome sets of octoploid strawberry, however, are more controversial[21-23,53-58,79,81]. These two sets of chromosomes were recognized by both Sargent et al.[56] and Tennessen et al.[57] as (1) closer to each other than to I or V, and (2) closer to I than to V, but have been differently partitioned into two subgenomes by several groups[21-23,53-58,79,81], typically based on their similarity to *F. iinumae*. This was recognized as a weak criterion, particularly if these two subgenomes are sister to one another, and so phylogenetically equidistant from *F. iinumae*, as suggested by Tennessen et al.[57]. Similarly, Sargent et al.[56] identified a set of genetic markers (their NN) that are predominantly found on what they provisionally referred to as X1/X2 subgenomes (i.e., the fourteen non-I, non-V) chromosomes. Edger et al.[58] partitioned these fourteen chromosomes into two sets based on their protein-coding similarity to *F. nipponica* and *F. viridis*[22,58], which has been questioned by Liston et al.[21] and Feng et al.[23]. We note that

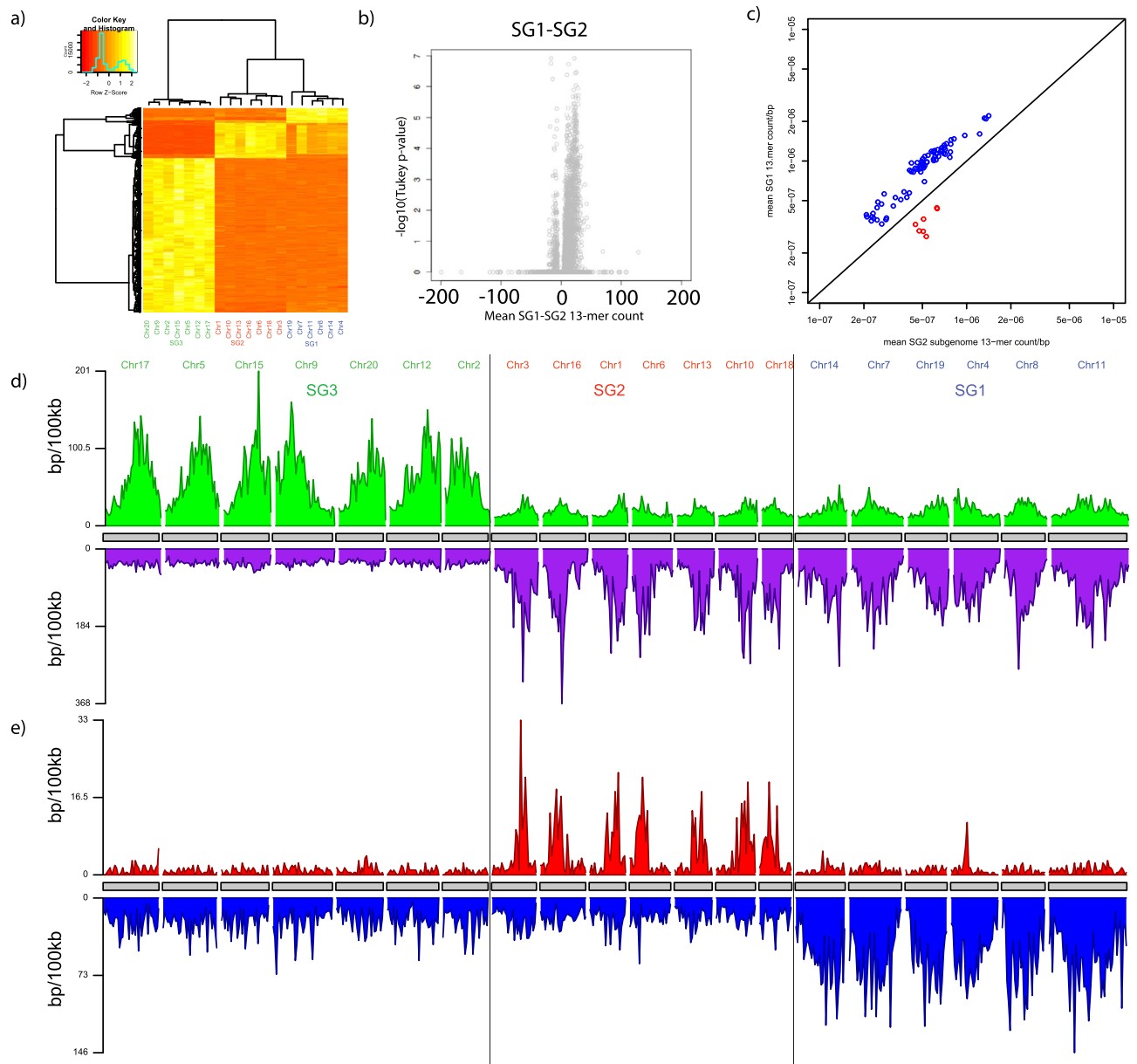

**Fig. 4 | Camelina allohexaploidy.** *Camelina sativa* is an allohexaploid whose ancestry has been inferred by comparison with extant diploids. We sought to identify subgenomes using the chromosomal distribution of k-mers in the hexaploid genome of *C. sativa* without reference to external data (Supplementary Note 4). **a** Heatmap showing 13-mer density as a function of *C. sativa* chromosome number (columns, clustered on top) vs. 13-mers found to differentiate subgenomes (rows, clustered on left), which are consistent with sub-genome assignments from comparisons with diploids[20]. SG1, SG2, and SG3 chromosomes are labeled in blue, red, and green, respectively. **b** Volcano plot showing Bonferroni-corrected Tukey *p*-value (Bonferroni-corrected; df = 17) vs. mean 13-mer count difference SG1-SG2 *Camelina* subgenomes. Each point is a 13mer. Tukey's HSD test is one-sided. Effect

size is shown on the *x*-axis (converted to 13mer count/chromosome) and 95% Confidence Intervals for each 13mer can be found in Supplementary Data 4. **c** Scatterplot showing mean SG1-vs-SG2 13-mer count/bp, with SG1- and SG2-enriched 13-mers shown in blue, and red, respectively. Black line is *y* = *x*. **d** Karyogram showing density of SG3-enriched (above, green) and SG1 + SG2-enriched (below, purple) 13-mers in 100-kb bins along each of the chromosomes of *C. sativa*. **e** Similar to (**d**) but contrasting density of 13-mers enriched in SG1 (blue, below) and SG2 (red, above). In both (**d**) and (**e**), chromosomes are labeled according to their subgenome assignment. Source data are provided as a Source Data file.

Edger et al.[58] proposal of extensive homoeologous exchange in octo-ploid strawberry would further confound the partitioning of these chromosome sets based on comparison with diploid relatives.

Our method using shared repetitive content, however, readily partitions these difficult-to-assign chromosomes into distinct sub-genomes, which we designate T1 and T2 (Fig. 5). This partitioning is different from previously proposed subgenome hypotheses[21–23,53–58,79,81]. As suggested by Tennessen et al. protein-coding analysis[57] and Sargent et al. haplo-SNP analysis[56] these two subgenomes are related to each

other and to I based on sharing of 13-mers between T1 and T2, and among all three of T1, T2, and I. Furthermore, T1 and T2 are differentiated from one another by a set of characteristic k-mers that are enriched in T1 relative to T2 (Fig. 5; Supplementary Note 5-7; Supplementary Fig. 6-8). Applying ANOVA to the normalized counts per chromosome of 423,429 13-mers that occur at least $N_{min} = 100$ times across the genome we identify 92 13-mers that support our T1-T2 partition with a density of at least 1 kb/Mb, with 91 13-mers enriched in T1 with a density of at least 1 kb/Mb and 1 enriched in T2

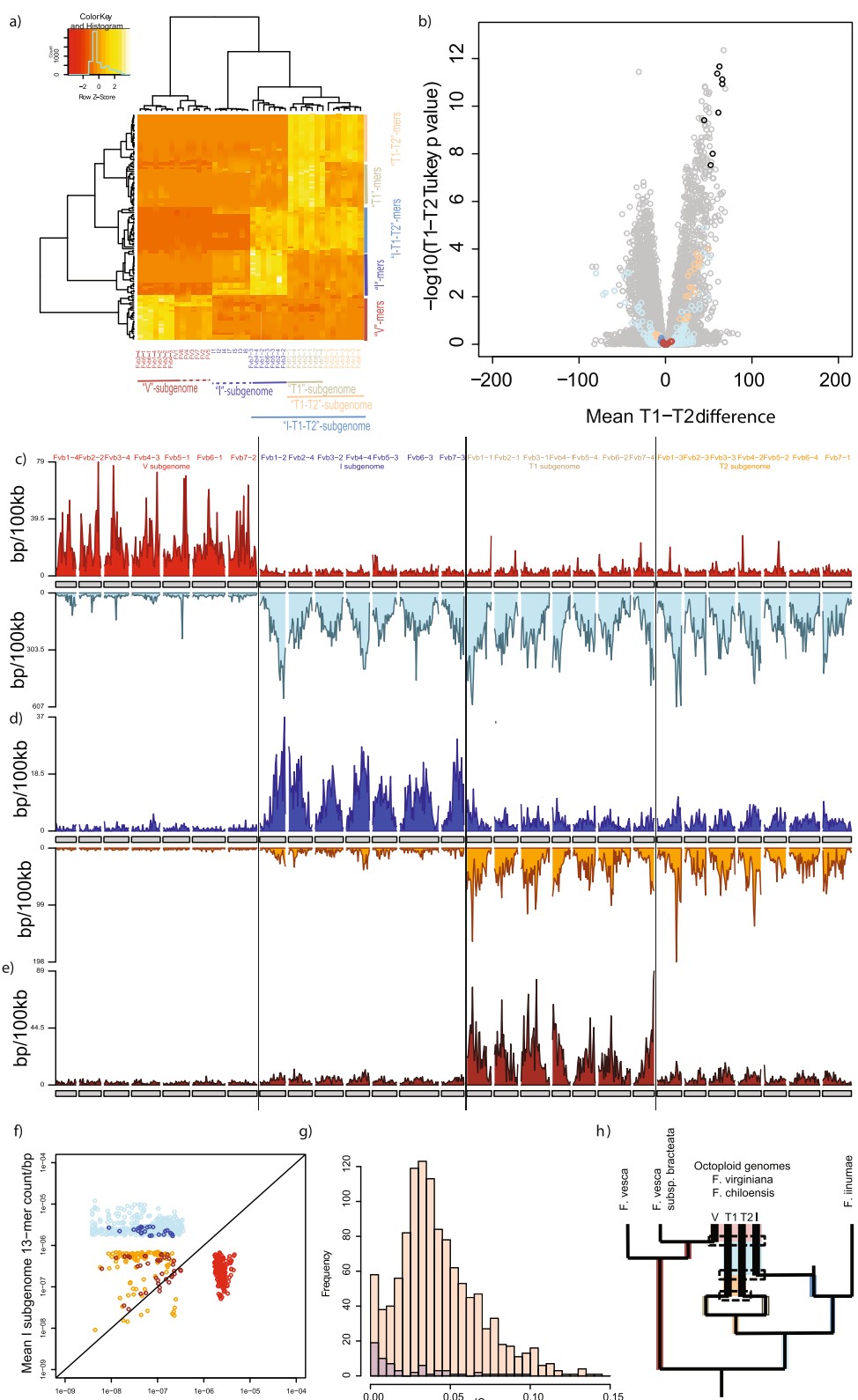

(Bonferroni-corrected significance threshold of 0.05; Tukey's HSD; df = 24). Similarly, 545 13-mers support the partition of I relative to T1 and T2 and 4,020 13-mers support the partition of the V subgenome from I, T1, and T2. The distributions of these 13-mers along each chromosome are shown in Fig. 5c–e.

Using the ANOVA/Tukey's range test approach, we can also test the specific 'nipponica'/'viridis' subgenome hypothesis proposed by Edger et al.[22,58]. Instead of assessing each k-mer for a difference in mean T1 vs. T2 density (Fig. 5b), we assessed all k-mers for significant differences in density between the hypothesized 'nipponica' and 'viridis' groupings (Supplementary Fig. 6b; Supplementary Note 6). We did not, however, find any significantly enriched 13-mer repetitive markers that support this proposed 'nipponica'/'viridis' subgenome partition (again using a Bonferroni-corrected threshold of $p < 0.05$; Tukey's

**Fig. 5 | Repetitive sequences partition the octoploid strawberry genome into distinct subgenomes. a** Heatmap showing chromosomal clustering based on k-mers that are enriched in one or more chromosomes of each homeologous quartet. Chromosomes (columns) are labeled according to Edger et al.[13]. The diploid *F. vesca* and *F. iinumae* genomes are included to show that they share 13mers with the V and I subgenomes respectively. The visualization uses 25 13-mers of each enrichment type. Full dataset in Supplementary Fig. 7. **b** Volcano plot showing Bonferroni-corrected Tukey *p*-value (df = 24) vs. mean 13-mer count difference between T1-T2 subgenomes. Black dots correspond to T1-enriched 13mers, orange to T1-T2, light blue to I-T1-T2, dark blue to I, and red to V. Tukey's test is one-sided. Effect size is shown on the *x*-axis and 95% Confidence Intervals for each 13mer can be found in Supplementary Data 4. **c** Karyogram of V-enriched 13-mer density in 100-kb bins (red, above chromosomes), and I/T1/T2 enriched 13-mer density (light blue, below). **d** Karyogram of I-enriched 13-mer density in 100-kb bins (dark blue, above chromosomes), and I/T1/T2 enriched 13-mer density (orange, below). **e** Karyogram of T1 enriched 13-mer density in 100-kb bins (brown, above chromosomes). In **c**–**e**, chromosome names colored as same in (**a**) and subgenomes are separated by lines. **f** Scatterplot showing mean I-subgenome vs. V-subgenome 13-mer count/bp. Only subgenome enriched 13-mers are shown, colored as in (**a**). **g** Histograms of Jukes-Cantor distance between 5′ and 3′ LTRs for retrotransposons in large families with I-T1-T2-enriched 13-mers. LTRs from I-T1-T2 families on I-T1-T2 chromosomes shown in red, and LTRs on V chromosomes shown in blue. The peak at ~0.035 (corresponding to ~3 mya, Supplementary Note 7) implies coexistence of subgenomes at that time. **h** Scenario for the evolution of octoploid strawberry from diploid progenitors, showing progressive addition of subgenomes and resulting intervals of shared transposon activity. Inferred polyploid ancestors include a tetraploid containing the ancestors of the T1 and T2 subgenomes (orange colored box); a hexaploid that adds the I subgenome to this tetraploid (blue box), and finally the formation of the octoploid by addition of the V subgenome to the hexaploid (red box). Source data are provided as a Source Data file.

HSD; df = 24). In contrast we found a strong signal differentiating the T1- and T2-subgenomes proposed here. We find these signal even using the non-parametric Dunn's test (Supplementary Fig. 8). Raw data and statistics for all 13-mers for strawberry analysis are provided in Supplementary Data 4 and 12.

## Dating the subgenome-specific retrotransposon activity
The grouping of subgenomes I, T1, and T2 shown in Fig. 5 and Supplementary Fig. 6-8 arises from recent shared transposon activity. Specifically, the 13-mers we define overlap with annotated families of long-terminal-repeat (LTR) retrotransposons in the octoploid genome assembly[58]. We inferred the timing of this activity from the sequence divergence between 5′- and 3′-LTRs [37,38]. The divergence of the crown group *Fragaria* has been estimated to be ~8 mya[83] (Supplementary Note 7). The extensive shared I-T1-T2 activity peaks at a sequence divergence corresponding to 3 mya (Fig. 5g). This in turn suggests that these three subgenomes were united in a hexaploid prior to the merger with the V subgenome. Combining these data with results from Liu et al.[55], Liston et al.[21], and Feng et al.[23] yields the scenario for the evolution of octoploid strawberry from V, I, T1, and T2 progenitors shown schematically in Fig. 5h. This suggests a longer shared history between the I, T1, and T2 subgenomes, due to their shared history as progenitors (diploid, tetraploid, or hexaploid) in our model. We note that the accelerated rate of sequence change in octoploid relative to diploid strawberries, coupled with possible extinction of relevant diploids, suggest possible explanations for the failure of protein-coding phylogeny-based methods to identify the T1 and T2 subgenomes (Supplementary Note 5–7). Strawberry retrotransposons and list of families are in Supplementary Data 2, while alignments are included in Supplementary Data 5.

## Allotetraploid tobacco and mixed chromosome ancestry
Although genome duplication after interspecific hybridization sets up an allotetraploid for disomic inheritance through consistent meiotic pairing of chromosomes from the same progenitor homeologous exchanges or replacements have been observed in known allotetraploids [12,18], and other chromosomal rearrangements can also occur [14,31]. Homoeologous exchanges and replacements in particular blur the distinction between subgenomes, and over time extensive homoeologous recombination would destroy the evolutionary coherence (shared ancestry) of chromosomes belonging to a subgenome. In order to test for possible exchanges between subgenomes, we developed a Hidden Markov Model whose hidden states are subgenome identity and emitted signals are local k-mer frequency in 100-kb bins along the genome (Supplementary Note 8), generalizing the method first described in Mitros et al.[18]. Homoeologous exchanges, or intersubgenome rearrangement, appear in this analysis as transitions in the subgenome state along a chromosome.

As an example of how subgenome-specific k-mers can be used to detect inter-subgenome rearrangements, we considered allotetraploid tobacco, *Nicotiana tabacum* (2*n* = 4*x* = 48) which has been investigated previously by Edwards et al. using comparisons with the diploid progenitor species *N. sylvestris* (S-subgenome) and *N. tomentosiformis* (T-subgenome) (Supplementary Note 8; Supplementary Fig. 9)[31]. We bootstrapped the discovery of subgenome-enriched k-mers by considering six homoeologous chromosome pairs of *N. tabacum* that showed no evidence for inter-subgenome exchange. The discovery of subgenome-enriched k-mers using a restricted set of homoeologous pairs parallels the use of specific chromosome triplets in *C. sativa* and the restriction to 24 out of the 25 chromosome pairs in goldfish. In this way we identified 13,447 and 11,655 13-mers enriched on the S- and T-subgenomes, respectively (Supplementary Note 8). Supplementary Fig. 9a shows that four chromosomes (Nt18, Nt22, Nt17, and Nt21 labeled in black) are less clearly assigned to subgenomes based on clustering at the whole chromosome scale.

These subgenome-enriched k-mers were in turn were used to define a Hidden Markov Model (HMM) to call 100 kilobase segments of the *N. tabacum* genome as either S- or T-like according to the k-mers that they contain (Supplementary Note 8). Supplementary Fig. 9e shows that *N. tabacum* chromosomes can be segmented into S- and T-like regions by HMM, and that the four outlier chromosomes include large exchanged or translocated regions that often occur at or near chromosome ends; several other chromosomes also show terminal exchanged regions (Supplementary Fig. 9d, e). These findings, made only with reference to the tetraploid *N. tabacum* genome, are consistent with the observations of Edwards et al. comparing *N. tabacum* to the diploids *N. sylvestris* and *N. tomentosiformis*[31].

We did not find notable homoeologous exchanges in the other allopolyploid genomes analyzed here. Specifically, since we found no evidence for large-scale segmental homoeologous exchange in strawberry, we suggest that the signals of homoeologous exchange proposed by Edger et al.[58] can be explained as arising from incomplete lineage sorting as observed in diploid species of *Fragaria*[23], rather than bona fide homoeologous exchange. Raw data and statistics for all 13-mers for tobacco analysis are provided in Supplementary Data 4 and 13. Tobacco retrotransposons and list of families are in Supplementary Data 3, while alignments are included in Supplementary Data 5.

## Biased fractionation and subgenome dominance
The subgenomes of many older polyploids have evolved asymmetrically [12,27,29,84–86]. "Dominant" and "submissive" subgenomes have been described based on differential gene loss or biased fractionation [29,85,87], differential gene expression [69,86,88], substitution rate[17], and insertions and deletions [69,85,86]. In particular, the biased fractionation of cotton, goldfish, and carp subgenomes has been extensively documented [28,61]. In allotetraploids, such differences

between subgenomes must have evolved after (or in association with) polyploid formation, since at the time of hybridization each progenitor contributes a complete gene set. Differential subgenome evolution may be influenced by intrinsic asymmetries between subgenomes that could affect subsequent gene retention/loss or expression changes.

Higher polyploids, however, may arise by the hybridization of progenitors of differing ploidy. For example, allohexaploid *C. sativa* was formed by combining an H[7]-like diploid with an N[6]N[7]-like tetraploid; similarly, the final stage in octoploid *F.* x *ananassa* formation was the combination of a V-like diploid with an I-T$_1$-T$_2$ hexaploid. In both cases, a diploid is combined with a pre-existing polyploid. Importantly, polyploids generally evolve under reduced purifying selection due to global genic redundancy, which leads to gene loss and degradation of gene expression.

Thus, while a diploid progenitor of a higher polyploid always contributes a complete gene set at the time of hybridization, polyploid progenitors (e.g., tetraploid for *C. sativa*, hexaploid for *F.* x *ananassa*) are typically already partially degraded by gene loss and/or diminished gene expression. For example, in strawberry the I-T$_1$-T$_2$ subgenomes had already evolved under millions of years of redundancy within the hexaploid progenitor prior to hybridization with the diploid V progenitor; at the time of hybridization, however, the V genome was intact (Fig. 5f). It follows that the V subgenome of contemporary octoploid strawberry is expected to have suffered less disruption (due to gene loss or altered expression) than its I-T$_1$-T$_2$ counterparts simply based on timing.

It follows from these general considerations that in a higher polyploid the most recently added subgenome should (1) possess higher gene retention (biased fractionation) since the other, polyploid, progenitor will already have lost redundant genes, and (2) more robust gene expression (genome dominance) since the other, polyploid, progenitor will have reduced gene expression of remaining redundant genes than the diploid progenitor. These evolutionary arguments are consistent with the finding that SG3 is dominant in *C. sativa*[20] and V is dominant in *F.* x *ananassa*[58], since these are the most recently added subgenomes. From this perspective, biased fractionation and subgenome expression dominance are expected in higher polyploids simply as a consequence of initial conditions without needing to appeal to intrinsic features of the sort that may drive subsequent asymmetric subgenome evolution in allotetraploids[89].

## Limitations of the k-mer approach

While we have shown that our method can be used to differentiate subgenomes in diverse cases, it has limitations. In particular, our logic relies on (1) exclusive TE activity during the period in which the progenitors are evolving separately (e.g., the lightning bolts of Figs. 1c), and (2) our ability to detect relicts of this activity in the polyploid genome by enrichment of k-mer counts. Condition (1) may be violated when the two progenitors of an allotetraploid are so closely related that they have not had time to develop distinct TE complements. Such a case may also be difficult to distinguish from autotetraploidy, especially if the progenitors are so closely related that they can pair and recombine, erasing any initial differentiation between subgenomes. Tetraploid *Arabidopsis suecica*[90] appears to be a marginal case where the two subgenomes are readily differentiated by the collection of subgenome-enriched 13-mers using hierarchical clustering (Supplementary Fig. 10a; Supplementary Note 9). In this case, however, none of the subgenome-enriched 13-mers are statistically significant individually, suggesting that our *p*-values may be overly conservative.

Condition (2) may be violated by polyploids whose hybridization occurred long enough ago that subsequent mutation and/or genomic turnover has obscured or erased the evidence for progenitor-specific TE activity. Thus, for example, we do not find a significant subgenome-specific k-mer signal in *Brassica rapa*, a paleo-hexaploid that is thought to have arisen more than 6 mya[14] (Supplementary Fig. 10b; Supplementary Note 10). This time period is significant because it is older than the average half-life of transposable elements in grass (and likely other plant) genomes[38], although we cannot rule out the possibility that the diploid progenitors had not diverged sufficiently to allow expansion of specific TEs as required for our method to detect subgenomes. The time-scale over which TE relicts can persist in a genome, however, appears to be lineage-specific, since subgenome-specific DNA transposons (Harbingers) were identified in *Xenopus laevis* that were active more than 15 mya[17]. Raw data and statistics for all 13-mers for *A. suecica* analysis are provided in Supplementary Data 4 and 14.

## Relationship of k-mers to transposable element evolution

Our approach relies on past progenitor-specific bursts of TE activity that distributed multiple copies across all chromosomes of the progenitor's genome (lightning bolts in Fig. 1). These copies must be (1) recognizable in the polyploid genome, and (2) differentiable at the sequence level from other, possibly similar, TEs that were active either before the progenitors diverged or after allopolyploid formation. In practice this often means we are looking at a specific subfamily of a larger TE family of related elements that may have a broader temporal and genomic distribution. To identify relevant TE subfamilies we can intersect the genomic positions of subgenome-specific k-mers with TE annotations of the genome, as described above for strawberry and tobacco. We note that the subgenome-specific k-mer footprint will typically only highlight a small portion of the TE sequence, since the k-mer analysis is focused specifically on those sequences that differentiate a progenitor-specific burst of activity from related elements that may have been active either before the divergence of progenitors or after polyploidization. In the case of LTR-retrotransposons, such as those used in the strawberry analysis, timing and phylogenetic relationships among surviving genomic copies can be characterized by comparing their long-terminal-repeat sequences, which themselves may or may not contain diagnostic k-mers.

Subgenome diagnostic k-mers/transposons tend to fall into one of two classes: (1) those that are truly exclusive to one subgenome due to novel activity in the corresponding progenitor, and (2) those that are show a highly asymmetric distribution between subgenomes, which can occur if the TE was already present in a common ancestor of more than one subgenome but became differentially active in only one progenitor after speciation but before hybridization (Fig. 1). Examples of these behaviors from the tetraploid tobacco genome are shown in Supplementary Fig. 11. As an example of a sub-genome-exclusive family, Supplementary Fig. 11a shows a dendrogram based on Jukes-Cantor distances between LTRs of tobacco retroelement family fam_150. This family was identified in the tetraploid tobacco genome using standard repeat identification software[44], without regard to k-mers or subgenomes, and is a member of the Sirevirus (Ty1-*copia* superfamily) based on alignment of its inner sequences to a retroelement consensus sequence database[91]. Fam_150 is found only on the T-subgenome; as expected this subfamily is marked by T-subgenome-specific k-mers. Alternatively, consider tobacco TE family fam_62, a member of the *tork* lineage of the Ty1-*copia* superfamily, as shown in Supplementary Fig. 11b,c. While fam_62 has members on both subgenomes, only one subclade showing subgenome bias. In this case, the diagnostic S-subgenome 13-mer falls in the LTR and a multiple alignment of this region is shown in Supplementary Fig. 11c. Specifically, we find a T > C transition specific to the S-subgenome, with T representing the ancestral state; the T > C mutation occurred before or during the expansion of this S-subgenome-specific element. Only this specific 13-mer passed our stringent multiple-test-corrected *p*-value threshold. We investigated the other 13-mers overlapping this site and found that

they often had strong raw *p*-values (Tukey's HSD test, df = 22; *p* < 1e-6; Supplementary Data 4) but were not significant after Bonferroni correction (*n* = 838,357 k-mers tested in tobacco) suggesting that our analysis may be overly conservative.

Taken together, these results show our methodology for identifying subgenomes in allopolyploids is robust in diverse plant and animal systems. We identify a clustering of chromosomes for allo-octoploid *Fragaria* x *ananassa* subgenomes that is consistent with previous comparisons to related diploids. In addition, the shared repetitive signal between subgenomes provides a positive signal to know the order in which the progenitor diploids were hybridized into the higher ploidy organism. Our k-mer analyses are intrinsic to the allopolyploid, allowing for easier identification of subgenomes without needing to sample the genomes of diploid or other candidate progenitors. These results are important for understanding how we expect the genomes of many crops and model organisms to evolve and highlight the need to develop rigorous methodological approaches to study polyploid genomes. Computational tools for applying this method to other polyploids are provided at our Github (https://github.com/amsession/Kmer-based-Subgenome-Mapping).

## Methods

### Genome versions and custom code
All genome versions and sources for fasta files are included in Supplementary Table 1. Our entire algorithm, with scripts generalized for different orders of polyploids(up to octoploid), is available at Github (https://github.com/amsession/Kmer-based-Subgenome-Mapping). Any specific modifications to the algorithm for a specific species (such as only using a subset of chromosomes to start in *Camelina* and *Nicotiana*) are detailed in Supplementary Note 4 and 8.

### Partitioning chromosomes into subgenomes
Generally, we parsed genome sequence files into fasta format using the database management tools of BLAST + [92] and used Jellyfish[93] to count k-mers on each chromosome (typically with *k* = 13). We used the canonical flag to group together sequences with their reverse complement, since strand information is not relevant. We use R[94] to normalize the k-mer count by chromosome length, generating the 13-mer count/bp measure for statistical analysis, which generally produces normal distributions at the subgenome level after log transformation (Supplementary Fig. 1). For all analyses in this paper, we used $\log_{10}$. For each homoeologous chromosome pair, we look for k-mers that are enriched at least two-fold, then take the intersection of 13-mers that differentiate all pairs (unless otherwise noted, such as for Goldfish). Chromosomes are clustered hierarchically using 1-*r* as the distance between chromosomes (clustered at top of canonical heatmap, shown in Figs. 2–5) or 13-mers (clustered to the left of heatmap), where *r* is the Pearson correlation coefficient. Heatmaps are generated by the heatmap.2 function from the gplots package, setting the scale to each row (k-mer). Similar to the chromosomes, k-mers are clustered hierarchically using 1-*r* as the distance. Visual inspection of the heatmaps is used to complete the subgenome assignments before proceeding to statistical validation.

### Statistical analysis
We use the aov and TukeyHSD functions in base R to generate a table of ANOVA results and *p*-values for each subgenome comparison. The assumptions of ANOVA are satisfied with our data, (1) the measurement of 13-mer counts are independent, (2) the subgenome distributions are approximately normal (Supplementary Fig. 1), and (3) within each species, the distributions have roughly equal variance. Tukey's HSD test allows us to assign a *p*-value for each k-mer's ability to differentiate subgenomes. Bonferroni corrections are done with the p.adjust function in base R. For each k-mer that meets the minimum count in each species, we provide a table that has the ANOVA and

Tukey's HSD statistics, including ANOVA F-statistic, Tukey *p*-values, effect sizes, and 95% Confidence Intervals (Supplementary Data 4). For *Camelina* and strawberry multiple columns are provided to show the Tukey statistics for each pairwise subgenome comparison. By definition, Tukey's HSD test is a one-sided test.

### Hidden Markov Model and visualization of k-mer density
We use BLASTN to align significant 13-mers to the genome and determine their locations, followed by bedtools[95] to merge adjacent or overlapping 13-mer sequences. Karyograms show the density of the merged.bed files and are generated using the karyoploteR package[96]. We use bedtools to generate tables of 13-mer density/100-kb bin for use in the Hidden Markov Model step. The HMM and Viterbi path are generated by the HMM[97] package, and visualized using the karyoploteR package. Computational tools for applying this method to other polyploids are available at Github (https://github.com/amsession/Kmer-based-Subgenome-Mapping).

### Intact retrotransposon annotations and subfamily building
We used LTR-Harvest[44] to annotate intact retrotransposons in *Camelina*, strawberry, and tobacco using default parameters. We named each retrotransposon by its genomic location (i.e Chr1:1-100), and created fasta files for the LTRs and inner sequences separately using BLAST + . Subgenome-enriched 13-mers were assigned to retrotransposons based on overlap of genomic coordinates. We aligned LTRs to one another to build subfamilies separately from subgenome assignments. An all vs. all BLAST was done for the LTRs of each species with e value cutoff of $1e^{-2}$. Alignment across 90% of the length of both the query and subject was accepted as full-length evidence of similarity.

### Assignment of retrotransposons to larger families
Retrotransposons were assigned to larger families by best-hit alignment to GyDB[91] consensi. *Pol* peptide cosensi for retrotransposon families were obtained from the GyDB website (https://gydb.org). The peptide consensi were aligned to the inner sequences described above by tblastn, using 1e-10 as the e value cutoff. We note that many of the inner sequences are assembled as N's and cannot be assigned. This is to be expected due to the high degree of similarity of these regions between retrotransposons that may have different LTR sequences. BLAST bit score was used to determine the best hit for each of the proteins in each retrotransposon.

### Reporting summary
Further information on research design is available in the Nature Portfolio Reporting Summary linked to this article.

## Data availability
Data supporting the findings of this work are available within the paper and its Supplementary Information files. A reporting summary for this article is available as a Supplementary Information file. The genome data sets used in this study were obtained from publicly available databases. The exact genome version and source for each species are in Supplementary Table 1. Source data are provided with this paper.

## Code availability
Custom scripts for applying the methods described in this manuscript are available at Github [https://github.com/amsession/Kmer-based-Subgenome-Mapping].

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

## Acknowledgements

Work conducted at the U.S. Department of Energy Joint Genome Institute, a DOE Office of Science User Facility, is supported by the Office of Science of the U.S. Department of Energy under Contract No. DE-AC02-05CH11231. Additional support was provided by the DOE Center for Advanced Bioenergy and Bioproducts Innovation, which is supported by the U.S. Department of Energy, Office of Science, and Office of Biological and Environmental Research under Award Number DE-SC0018420. Any opinions, findings, and conclusions or recommendations expressed in this publication are those of the author(s) and do not necessarily reflect the views of the U.S. Department of Energy. D.S.R. is grateful for support from the Chan-Zuckerberg BioHub and the Marthella Foskett Brown family. We thank Jarrod Chapman for valuable early discussions about k-mer distributions in allotetraploids and Robert Dalipovic for comments on the manuscript.

## Author contributions

A.M.S. performed the analysis and helped write the manuscript. D.S.R. provided project leadership and helped write the manuscript.

## Competing interests

The authors declare no competing interests.
