## [Peer Review File · Nature Communications]

Transposon signatures of allopolyploid subgenome evolutionReviewers' Comments:

Reviewer #1:

Remarks to the Author:

Review: Session and Rokhsar

The authors describe and demonstrate an innovative new method for defining subgenome compositions in allopolyploids. The method relies upon the identification of k-mers that display varying and informative levels of subgenome-specific enrichment. The method is validated via subgenome analysis of two well-studied allopolyploids, the outcomes of which conformed to expectations based on prior knowledge. The method is then used to clarify the subgenome composition of an octoploid strawberry, thereby resolving a controversy precipitated by the claims of Edger et al (2019).

This paper is very clearly written, and requires very little technical revision, with the few exceptions being noted below. However, the version I received was lacking in supporting data of two types (see below), and in my view these substantive omissions must be rectified to make the paper suitable for publication. I also suggest some additional citations of prior work, and attention to several small but substantive points.

Substantive revisions:

Line 137: The authors propose the general equation $2n=2px$ to specify genome composition, where p is an indicator of ploidy ($p = 1$ for diploids, 2 for tetraploids, etc), and x is the basic chromosome number. However, this form of the equation works only when all subgenomes have the same value of x . However, later (lines 244-245) the authors report that the subgenomes of hexaploid *C. sativa* have x values of 6, 7, and 7. How would the authors modify their general equation to account for such an instance?

Lines 194-195. I am not convinced that the term "phylogenetic analysis" applies to Figure 1b. Is this figure really the result of an analysis, or is it simply a depiction of one theoretical case in point?

Technical revisions:

Figure 1: this figure needs more explanation. First, it takes a moment to realize that the information flow is from bottom to top (especially given that the only arrows in the figure point downward). It would add clarity to indicate that, in each figure, the bottom chromosome set is that of the diploid ancestor, where $x=3$. Also, the intent of the differing box (chromosome) colors is not immediately clear: for instance, in figure A the box colors all seem to be the same, thus suggesting auto- rather than allo-polyploidy. Also, what do the two jagged "lightning bolts" indicate in part c? Transposition episodes? If so, why not state it for the sake of clarity.

Figure 5f: Some additional explanatory information is needed in the legend of this figure. Specifically, what do the three colored boxes indicate? I can guess, but why not make the color code meaning explicit?

Line 56: insert space between "controversial" and "in".

Line 218: delete first "the"

Line 239: insert "came from" after "that"

Line 323: Should the cited figure be 5e (not 1e)?

Overall organization. The Results section is an interweaving of results and discussion, while the brief Discussion section is better characterized as a Conclusions section. I suggest changing the section headings accordingly: "Results" becomes "Results and Discussion", and "Discussion" becomes "Conclusions".

Substantive revisions

K-mer lists. A key element of the results is the identification and utilization of informative k-mers in each of the studied systems. However, the materials provided to me for review did not include any listing of k-mers. From my perspective, the presentation of the k-mer lists is a requisite component of the results section, without which I would not regard the paper as publishable. Upon my request, a spreadsheet containing the k-mer lists for each study was provided to me, and inclusion of these lists as supplementary data would satisfy my concerns on this point.

K-mers and LTR families (lines 318-320, Suppl. Notes 5, and elsewhere). The authors state that they have defined overlap of 13mers with annotated families of LTR lists. However, they provide no specific

evidence or instances of this overlap, and I could not find any listing of the retrotransposon families in question within the materials I obtained for review. One solution to this would be to add a column of available transposon family identifications to the k-mer spreadsheets that were requested above. In my view, inclusion of detail about LTR families is a prerequisite for publication, unless the authors are in the process of presenting this information elsewhere in the very near future.

Line 263: The authors refer to the genome assembly project of Edger et al as a "genomic tour de force". This characterization seems fawning and overly reverential, especially given that this assembly is not consistently admired in the strawberry genomics community. The paper would be improved by its removal.

Suggested citations

The authors have cited Sargent et al (2016) (reference #61), but not to the extent warranted.

Sargent et al (2016) proposed a polyploid origins scenario for the octoploid *Fragaria* that is highly relevant to the model proposed by the present authors on lines 323-325 and 368-373, and Figure 5f.

Relevant papers not cited:

Yang Y and Davis TM. 2017. A complex phylogeny of *Fragaria* (strawberry) species is revealed by large-scale multi-locus analysis. *Genome Biology and Evolution*. 2017;9 (12) :3433-3448.

<https://doi.org/10.1093/gbe/evx214>

Liu B, Poulsen EG, Davis TM. 2016. Insight into octoploid strawberry (*Fragaria*) subgenome composition revealed by GISH analysis of pentaploid hybrids. *Genome* 59(2): 79-86,

<https://doi.org/10.1139/gen-2015-0116>

Reviewer #2:

Remarks to the Author:

The authors present a clever approach to identify which chromosomes in an allopolyploid derive from which progenitor lineage, based on transposon signatures. The basic idea of using transposon signatures to identify chromosome sets with shared ancestry was previously used by the same authors (among others) on several allotetraploids. Here, the authors provide a more statistically founded version of their method, and extend its applicability to higher-order allopolyploids. One particularly attractive feature of the methodology is that it can also reveal the relative timing of the events giving rise to higher polyploids.

The main focus of this paper is methodological, but the authors also present several case studies leading to biological insights that are interesting in their own right. The application of the method to octoploid strawberry for instance generates results that falsify the 'nipponica/viridis' chromosome grouping put forward earlier by Edger et al. (*Nature Genetics* 52(1):5-7). The authors also make a valid point that investigations on subgenome dominance in higher polyploids should take into account the order of hybridization events that gave rise to the polyploid, as some of the subgenomes may have experienced relaxed purifying selection longer than others.

I only have one major comment, and that is that a software package or script implementing the methodology outlined in this manuscript would be of great value to the allopolyploid community. Most of the separate steps involved (definition of homoeologous chromosome sets, k-mer counting, ANOVA/Tukey tests, clustering, visualization) are rather standard and easily implemented, but making the code available would definitely save others some time in re-implementing the authors' methodology, and facilitate its more general adoption in the community.

Minor comments :

- Meaning of colors in Figure 2b is not entirely clear, do these refer to 1 k-mer or a set/cluster of k-mers ?
- Supplementary Note 1 : '... divided into $x=25$ sets of homoeologous pairs.' should be '... divided into $n=25$ sets of homoeologous pairs.'

- line 209 : 'Extended Data Fig. 3c' should be 'Extended Data Fig. 1c'.
- line 233 : 'Chaudary et al.' : reference missing.
- line 236 : I had some difficulty correctly interpreting the sentences 'Due to the chromosomal rearrangements in *C. sativa* documented by Kagale et al ., we first identified candidate subgenome-enriched 13-mers by considering three clearly paralogous chromosome trios ((Csa15, Csa19, Csa01); (Csa17, Csa14, Csa3); and (Csa04, Csa06, Cs09)). Candidate 13-mers that from this analysis then yielded a consistent partitioning of all chromosomes into three subgenomes'. How can chromosomes that were rearranged yield a consistent partitioning ? Shouldn't there be some signal differentiating some (rearranged) chromosome parts from others ? If so, these signals are not immediately apparent on Figure 4. It only became clear to me after reading the 'homoeologous exchange' section on page 9 that there may not be any major chromosomal rearrangements in *C. sativa*. The aforementioned sentences may therefore need to be reformulated.
- Figure 4 : in the legend for panels c and d, 'bottom is SG3' should be replaced by 'bottom is SG1'. In panel d, I guess the SG1/SG2 shared repeat density is plotted, not the SG2/SG3 shared repeat density.
- Supplementary note 2 : spell out 'EDF3' as Extended Data Figure 3.
- Figure 5 : the labeling of V-mers, I-mers... below the x-axis in panel a is counterintuitive, these labels should be on the y-axis. I think it is also worth explaining in the text why the 'V-mers' for instance are better able to distinguish the *F. vesca*-related subgenome of the octoploid than the *F. vesca* genome itself (differences in color intensity on Figure 5b). I'm assuming this is due to divergent transposon activity since their last common ancestor ?
- Figure 5 : reference in panel e legend to supplementary note 3 should be to supplementary note 5.
- line 316 : spell out '(EDF. 5g)'.
- Extended data figure 5 legend erroneously refers to Figure 1a.
- supplementary note 3, 4 and 5 : figure and note references in these sections are incorrect. A lot of the text in these sections is redundant with the main text.
- supplementary note 5 : Last two sentences of this supplementary note need to be reformulated (and references added).
- line 323 : reference to a non-existing Figure 1e.
- line 336 : 'share ancestry' should be 'shared ancestry'.
- several of the references contain formatting errors.

Reviewer #3:

Remarks to the Author:

In the paper entitled "Transposon signatures of allopolyploid subgenome evolution", the authors present a method to identify chromosome subgenomes in allopolyploid species. This approach is novel (though I do not follow the latest publications in the field) and can contribute to the analysis of subgenomes and their diploidization within allopolyploid animal and plant genomes. I question whether three examples are sufficiently demonstrating the robustness or applicability of the proposed approach. What is really needed is to spend some more time on explaining why these model groups were analyzed (not other). I failed to find any note on the applicability of the developed approach in old(er) polyploids, that is considerably diploidized genomes, such as "diploid" *Brassica* spp. (*B. rapa*). This should be discussed in the Discussion section and of course it would be great to include analysis of at least one of more diploidized genomes.

Introduction

It is incorrectly stated that the only pathway how allopolyploids are formed is hybridization followed by genome doubling. As the authors know an allopolyploid can be formed also in a single step or in a single step with preceding chromosome-set doubling in parental species.

P2, L56: Please correct and rephrase: "... to be particularly controversial in part due to...".

P3, L85: Figure 1. Please consider to clarify in the caption what the orange and blue flashes indicate.

Fig. 1c fails to show what is described in its legend ("progenitors accumulate unique complements of transposable elements").

P3, L105: "An important contrast is that since we do not typically have a training set of chromosomes of known provenance, however, without a training set of known authorship, we must bootstrap the identification of discriminatory DNA "words" from chromosome comparisons." Please rephrase this sentence as it is rather unclear and somewhat interrupts the otherwise quite clear narrative of this paragraph.

P3, L108: How k-mers were specific to transposable elements? Were genes masked on pseudo-chromosomes to identify only k-mer markers for TEs? Not specified in text here neither in Methods.

P4, L117-119: Here or elsewhere the authors should justify the choice of their allopolyploid genomes. Why not *Arabidopsis suecica* or *Tragopogon* spp. (for example)?

P4, L120: I am afraid that this entire sentence is messy and even repeated reading does not help to get some clear message. My understanding is that the authors confirm the existence of a tetraploid ancestral genome of *C. sativa*, but it is less clear what is meant by "missing diploid ancestor". This diploid ancestor was identified in the recently published (and cited) Plant Cell paper. Btw. what is SG217? I see now, that means SG2. By comparing the two cited papers, the Plant Cell paper gives much clearer answer on the parentage of the *C. sativa* genome than the G3 paper. This should be taken into account.

P4, L125: This paragraph should be re-written to set the scene properly. Using "controversial", "other observations" and "novel partitioning" without giving corresponding details makes this paragraph more or less useless. I was not able to conclude anything from this text. If the order how parental genomes "were added" into the 8x genome was inferred, did you also identify the (other) two subgenomes discussed? The last sentence is totally enigmatic (what is "neutral explanation"? *F. vesca* subgenome if *F. vesca* subgenomes are introduced above? What observations from allohexaploid *Camelina*?)

Methods

Origin of data not presented, which sequences/genome assembly versions were used? What was the source of transposable element sequences (and their distribution across the genomes)? Please add the information. Nothing is said about how the k-mer length was chosen. (Were other k-mer lengths tested?)

P5, L170: What tool was used for clustering?

P5, L173: What method did you choose? Hierarchical clustering or other methods?

P5, L188: No mention whether ANOVA assumptions were tested before using ANOVA.

Results

p. 5: I understand that that it is some sort of jargon (practical usage), however, calling genomes with 50 chromosome pairs as paleotraploids and those with 25 pairs as diploids seems to be funny, particularly in the paper focused on this phenomenon. Consider some short explanation or rewording. I think this entire paragraph has to be rewritten to ease its understanding. For example, I was not able to comprehend where from the A subgenome comes from, what are L and S subgenomes? As there is no useful figure accompanying this text, its content cannot be understood.

L225: " $2n=40$, $p=3$, $x\sim 7$, including one chromosome fusion)" I was not sure what is the meaning of "one chromosome fusion"?

l. 249 onwards: After checking the two cited papers, I conclude that the authors are intentionally bagatelizing these published results and conclusions, and over-emphasize their methods and results.

l. 255: I was not sure what I should imagine under "order of hybridization of *C. sativa* subgenomes that is intrinsic to the hexaploid itself" The order of hybridization (among subgenomes) is intrinsic to the hexaploid (genome)? The statement that "the previous finding of the existence of a related tetraploid to two of the subgenomes of the hexaploid merely implies ancestry, while our analysis provides a strong positive signal for the order..." is simply false. Both papers, mainly Mandakova et al., report on the order how subgenomes were hybridizing. In fact GISH-based identification of subgenomes (based on repeats) is congruent with your inference (fig. 4c, d). As for figure 4, it is

puzzling why the authors do not use chromosome IDs from the cited Plant Cell paper, as their graphic subgenome-chromosome arrangement would look more elegant than at present (fig. 4a).

p. 8, L297: please reword this to improve the clarity „but this has been questioned by Liston et al.18; the proposal of extensive homoeologous exchange by Edger et al.46,47 would further weaken this signal.”

P9: „We note that the accelerated rate of sequence change in octoploid relative to diploid strawberries, coupled with possible extinction of relevant diploids, suggest possible explanations for the failure of methods dependent on protein coding phylogeny to identify the T1 and T2 sub-genomes”. Perhaps I missed something but did you identify the two subgenomes? My understanding is that you identified two subgenomes (T1 and T2) being closer to I subgenome (than to V subgenome). Then, what is the difference to say “the failure of methods dependent on protein coding phylogeny”? What exactly is meant by “the failure”?

P: 9, L:345. “Such signals presumably do not arise in *Camelina* and the cyprinids because the interspecific divergences are more clearly defined.” I admit that this explanation is not clear to me. This argumentation is actually (to some extent) negating what was said about the two sub-genomes in the hexaploid *C. sativa* genome – all evidence (here and published) points to close genomic and phylogenomic relatedness among the two subgenomes.

p. 10, l.354: I do not understand the significance of this paragraph. This narrative hardly fits into Results. Although I value this argumentation of the authors, I was not able to realize their contribution to this matter. Did you document the dominance of SG3 or not? Similarly, I do not understand the meaning of the last sentence. Why do we need to know the order of hybridization events to define dominant subgenome(s)? My understanding, also based on herein proposed method, is that (sub)genome dominance should be inferred (in ideal case) blindly, without some a priori assumptions. Please clarify, modify the wording.

Also the next paragraph is confusing. I do not think that Mike Freeling, Jim Birchler and others somehow consider that genome (sub)dominance is independent of time and the order in which genomes merge in (allo)polyploids. It follows that I do not get how conclusions of Edger et al. differ from yours. I also do not fully understand what is meant by “neutral expectation”. What is “neutral” on expectation that redundant (duplicated) genes are more often lost or have modified expression? This paragraph is also more discussion than results.

Reviewer #4:
Remarks to the Author:
Review

This work by Session & Rokhsar identifies k-mers specific to sets of divergent chromosomes to distinguish divergent subgenomes in polyploids. This in silico version of molecular cytogenetics is an original way of looking at assemblies of complex genomes. Here, reanalysis of the conserved carp genome as well as dynamic plant genomes such as in *Camelina* and in the very complex *Fragaria* was used as a validation of the approach.

Not being the necessary expert of the “controversial” (sic?) *Fragaria* genome to ascertain to what extent outputs of the approach offers added-value, I can only report that results in that case as well as other case studies were convincing.

One of the main issues requesting clarification from my point of view is the extent to which tracked k-mers are derived from transposable elements. That the statistical toolbox used here can detect highly-repeated sub-genome markers is clear, but I have not read about the validation of k-mers being part of (retro?)transposons rather than e.g. tandem repeats (i.e. other than centromeric ones). On l. 319,

it is simply stated that "13mers we define overlap with annotated families of retrotransposons". This looks a critical validation for part of the results focused on transposons (e.g. dating). If a strict association (specificity) of tracked 13-mers with transposable elements is hardly apparent, I would suggest to skip most of related arguments (i.e. throughout).

In general, the presentation of the work could be improved. The core text reads well, although several lengthy descriptions and other typos in the spelling of figures and tables made it difficult to follow the details. Supplementary information is very rich and it is sometimes impossible to be confident about claimed justifications. I will not go beyond the effort of requesting a thorough check.

An a more minor side:

I would refrain to relate allopolyploids to permanent heterozygosity. It is "fixed heterozygosity" of a very different kind that found in e.g. *Oenothera* that has been termed "permanent".

Two kinds of polyploids is at best schematic and poorly accounts for increasingly diverged progenitors. Similarly over-simplistic: genome doubling does not necessarily occurs "after" hybridization at the origin of allopolyploids, aso...

Reviewer #1 (Remarks to the Author):

Review: Session and Rokhsar

The authors describe and demonstrate an innovative new method for defining subgenome compositions in allopolyploids. The method relies upon the identification of k-mers that display varying and informative levels of subgenome-specific enrichment. The method is validated via subgenome analysis of two well-studied allopolyploids, the outcomes of which conformed to expectations based on prior knowledge. The method is then used to clarify the subgenome composition of an octoploid strawberry, thereby resolving a controversy precipitated by the claims of Edger et al (2019).

This paper is very clearly written, and requires very little technical revision, with the few exceptions being noted below. However, the version I received was lacking in supporting data of two types (see below), and in my view these substantive omissions must be rectified to make the paper suitable for publication. I also suggest some additional citations of prior work, and attention to several small but substantive points.

Thanks for your helpful comments, which we think have clearly improved the manuscript. We have address the specific revisions below, and in particular have added both types of requested supporting data and additional citations.

Substantive revisions:

Line 137: The authors propose the general equation $2n=2px$ to specify genome composition, where p is an indicator of ploidy ($p = 1$ for diploids, 2 for tetraploids, etc), and x is the basic chromosome number. However, this form of the equation works only when all subgenomes have the same value of x . However, later (lines 244-245) the authors report that the subgenomes of hexaploid *C. sativa* have x values of 6, 7, and 7. How would the authors modify their general equation to account for such an instance?

The formula $2n = 2px$ only holds in the absence of other processes that change chromosome number. For *Camelina*, the base chromosome number is $x=7$. In *C. sativa*, however, the hexaploid has $n = 20 = 7+7+6$ since in one of the subgenomes two chromosomes have become fused. In this case the “ $3x$ ” in the formula really becomes $(x+x+x')$ where $x=7$ and $x'=6$. We have now noted this in the text. A strength of our k-mer based approach is that such “dysploidy” (changes in chromosome number by translocation or other rearrangement) are easily accounted for. This is now discussed in lines 166-167 and 315-317

Lines 194-195. I am not convinced that the term “phylogenetic analysis” applies to Figure 1b. Is this figure really the result of an analysis, or is it simply a depiction of one theoretical case in point?

Thanks. We have changed the wording to avoid confusion. Figure 1 is a schematic laying out scenarios for thinking about allotetraploidy. In panel 1b, a diploid (shown to the left) is related to one of the subgenomes. The orange circle marks the speciation between the diploid and the progenitor of one subgenome. This is now clarified in the legend. The relationship between the diploid and the subgenome is typically determined by phylogenetic analysis of protein-coding sequences.

Technical revisions:

Figure 1: this figure needs more explanation. First, it take a moment to realize that the information flow is from bottom to top (especially given that the only arrows in the figure point downward). It would add clarity to indicate that, in each figure, the bottom chromosome set is that of the diploid ancestor, where $x=3$. Also, the intent of the differing box (chromosome) colors is not immediately clear: for instance, in figure A the box colors all seem to be the same, thus suggesting auto- rather than allo-polyploidy. Also, what do the two jagged “lightning bolts” indicate in part c? Transposition episodes? If so, why not state it for the sake of clarity.

Thanks. We have clarified and streamlined the figure legend as requested, including a discussion of colors. In general, we used gray to indicate chromosomes whose subgenome was not known, and different colors to represent subgenomes.

Figure 5f: Some additional explanatory information is needed in the legend of this figure. Specifically, what do the three colored boxes indicate? I can guess, but why not make the color code meaning explicit?

We have explained the boxes in the revised legend.

Line 56: insert space between “controversial” and “in”.

Line 218: delete first “the”

Line 239: insert “came from” after “that”

Line 323: Should the cited figure be 5e (not 1e)?

Thanks, for catching these typos. They have been corrected in the revised manuscript.

Overall organization. The Results section is an interweaving of results and discussion, while the brief Discussion section is better characterized as a Conclusions section. I suggest changing the section headings accordingly: “Results” becomes “Results and Discussion”, and “Discussion” becomes “Conclusions”.

Thanks, we have made these suggested changes.

Substantive revisions

K-mer lists. A key element of the results is the identification and utilization of informative k-mers in each of the studied systems. However, the materials provided to me for review did not include any listing of k-mers. From my perspective, the presentation of the k-mer lists is a requisite component of the results section, without which I would not regard the paper as publishable. Upon my request, a spreadsheet containing the k-mer lists for each study was provided to me, and inclusion of these lists as supplementary data would satisfy my concerns on this point. K-mers and LTR families (lines 318-320, Suppl. Notes 5, and elsewhere). The authors state that they have defined overlap of 13mers with annotated families of LTR lists. However, they provide no specific evidence or instances of this overlap, and I could not find any listing of the retrotransposon families in question within the materials I obtained for review. One solution to this would be to add a column of available transposon family identifications to the k-mer spreadsheets that were requested above. In my view, inclusion of detail about LTR families is a prerequisite for publication, unless the authors are in the process of presenting this information elsewhere in the very near future.

Thanks. We agree that the specific k-mer lists and overlap with LTRs are essential to provide, and apologize for not including these initially. We now provide the subgenome-enriched k-mer lists and corresponding subgenome as Supplementary Files 1-9. Supplemental File 12 contains compressed files that show the full 13-mer x chromosome matrix for each species discussed. For each of the four species (goldfish, carp, *Camelina*, strawberry), and for the newly added *Gossypium hirsutum*, *G. barbadense*, *Brassica napus*, *Arabidopsis suecica* and *Nicotiana tabacum* (tobacco), these files include the sub-genome-specific k-mers and their subgenome assignment, as well as the complete counts of all 13-mers by chromosome. (The complete counts x chromosome files are large (gigabytes) and are therefore provided as a compressed TAR archive, with one table for each species.)

For strawberry and *Camelina* we now provide fasta files of intact retrotransposons and assignments to subgenomes where possible (Supplementary File 10 and Supplementary File 11). We computed these retrotransposons using LTR-Harvest and the reference genomes as described in Supplementary Notes 2 and 3. We have also added a discussion of the intact retrotransposon overlap of the *Camelina* 13-mers in the supplement.

Please note we have not been able to obtain the repeat files (repeatmasker or gff) for goldfish and carp; these are apparently not available from publicly available resources. Indeed, a strength for our approach is that the “annotation” of repetitive content is not necessary for separating sub-genomes.

Line 263: The authors refer to the genome assembly project of Edger et al as a “genomic tour de force”. This characterization seems fawning and overly reverential, especially given that this assembly is not consistently admired in the strawberry genomics community. The paper would be improved by its removal.

Thank you. We originally included this remark since, even though we disprove their sub-genome assignments, we did not want to suggest that Edger et al’s genome assembly itself was in error. We have eliminated this statement as requested.

Suggested citations

The authors have cited Sargent et al (2016) (reference #61), but not to the extent warranted. Sargent et al (2016) proposed a polyploid origins scenario for the octoploid *Fragaria* that is highly relevant to the model proposed by the present authors on lines 323-325 and 368-373, and Figure 5f.

Thanks for drawing our attention to this. We have added a brief discussion of the agreement between our scenario and the findings of Sargent et al. 2016. Briefly, using single nucleotide variation across *Fragaria* they identify two closely related subgenomes of octoploid strawberry, which they collectively call X1/X2, without attempting to partition them further into distinct subgenomes. Sargent et al. 2016 show that these chromosomes are more closely related to the I-subgenome. The X1/X2 chromosomes are the same as the non-I-or-V chromosomes of Edger et al. 2019, who further claim to separate them into separate ‘nipponica’ and ‘viridis’ subgenomes. Our model resolves the X1/X2 group differently than Edger et al. does, and explicitly refutes their ‘nipponica’ and ‘viridis’ subgenome partitions.

Relevant papers not cited:

Yang Y and Davis TM. 2017. A complex phylogeny of *Fragaria* (strawberry) species is revealed by large-scale multi-locus analysis. *Genome Biology and Evolution*. 2017;9 (12):3433-3448. <https://doi.org/10.1093/gbe/evx214>

Liu B, Poulsen EG, Davis TM. 2016. Insight into octoploid strawberry (*Fragaria*) subgenome composition revealed by GISH analysis of pentaploid hybrids. *Genome* 59(2): 79-86, <https://doi.org/10.1139/gen-2015-0116>

Thanks for catching these omissions. We have added these citations throughout the text as appropriate. (Yang et al. is #54, Liu et al. is #55.)

Reviewer #2 (Remarks to the Author):

The authors present a clever approach to identify which chromosomes in an allopolyploid derive from which progenitor lineage, based on transposon signatures. The basic idea of using transposon signatures to identify chromosome sets with shared ancestry was previously used by the same authors (among others) on several allotetraploids. Here, the authors provide a more statistically founded version of their method, and extend its applicability to higher-order allopolyploids. One particularly attractive feature of the methodology is that it can also reveal the relative timing of the events giving rise to higher polyploids.

The main focus of this paper is methodological, but the authors also present several case studies leading to biological insights that are interesting in their own right. The application of the method to octoploid strawberry for instance generates results that falsify the 'nipponica/viridis' chromosome grouping put forward earlier by Edger et al. (*Nature Genetics* 52(1):5–7). The authors also make a valid point that investigations on subgenome dominance in higher polyploids should take into account the order of hybridization events that gave rise to the polyploid, as some of the subgenomes may have experienced relaxed purifying selection longer than others.

I only have one major comment, and that is that a software package or script implementing the methodology outlined in this manuscript would be of great value to the allopolyploid community. Most of the separate steps involved (definition of homoeologous chromosome sets, k-mer counting, ANOVA/Tukey tests, clustering, visualization) are rather standard and easily implemented, but making the code available would definitely save others some time in re-implementing the authors' methodology, and facilitate its more general adoption in the community.

Thank you. To allow others to use our methods, we now provide a GitHub containing the basic perl and R scripts for usage on tetraploids, hexaploids, and octoploids. (<https://github.com/amsession/Kmer-based-Subgenome-Mapping>). In addition, we include a tutorial on how to perform the analysis on *Brassica napus*, a well-studied allotetraploid, whose results are included in Supplemental Note 1 and Extended Data Figure 1.

Minor comments :

- Meaning of colors in Figure 2b is not entirely clear, do these refer to 1 k-mer or a set/cluster of k-mers ?

We have clarified in the revised figure legend that this is an example of a hexaploid with $p=3$ and $x=4$. The colors in Figure 2b and 2c represent groups of k-mers that differentiate subgenomes, paralleling Figure 1. Figure 2b shows that the initial definition of putatively sub-genome-specific k-mers can be done purely from knowledge of homoeologous sets (horizontal boxes). Figure 2c shows that these k-mer markers can be used to group chromosomes into subgenomes (vertical boxes).

- Supplementary Note 1 : ‘... divided into $x=25$ sets of homoeologous pairs.’ should be ‘... divided into $n=25$ sets of homoeologous pairs.’

This is correct as written, since “x” labels both the base chromosome number and the number of homoeologous sets. The variable “n” labels the number of (meiotically) homologous chromosome pairs. Since carp and goldfish have $2n=4x=100$, $n=50$ and $x=25$. This is now Supplementary Note 3.

- line 209 : ‘Extended Data Fig. 3c’ should be ‘Extended Data Fig. 1c’.
- line 233 : ‘Chaudary et al.’ : reference missing.

Thanks for catching these typos.

- line 236 : I had some difficulty correctly interpreting the sentences ‘Due to the chromosomal rearrangements in *C. sativa* documented by Kagale et al ., we first identified candidate subgenome-enriched 13-mers by considering three clearly paralogous chromosome trios ((Csa15, Csa19, Csa01); (Csa17, Csa14, Csa3); and (Csa04, Csa06, Cs09)). Candidate 13-mers that from this analysis then yielded a consistent partitioning of all chromosomes into three subgenomes’. How can chromosomes that were rearranged yield a consistent partitioning ? Shouldn’t there be some signal differentiating some (rearranged) chromosome parts from others ? If so, these signals are not immediately apparent on Figure 4. It only became clear to me after reading the ‘homoeologous exchange’ section on page 9 that there may not be any major chromosomal rearrangements in *C. sativa*. The aforementioned sentences may therefore need to be reformulated.

Thanks we have reformulated this paragraph to make this (hopefully) clearer, lines 331-341. The chromosomes within each of these three triples are directly homoeologous without needing to consider any rearrangements. We identify 13-mers that consistently partition these nine chromosomes into three groups of common repetitive content, one from each homoeologous triple. These same 13-mers can be then used to assign the remaining 11 chromosomes to the three sub-genomes, adding 4, 4, and 3 chromosomes, respectively. The difference is accounted for by a chromosomal fusion, which we did not have to assume – it comes out of the analysis. Also, other rearrangements are seen to be intra-sub-genome rather than between sub-genome. This is consistent with the fact that our HMM analysis does not detect any homoeologous exchanges in *C. sativa*.

- Figure 4 : in the legend for panels c and d, ‘bottom is SG3’ should be replaced by ‘bottom is SG1’. In panel d, I guess the SG1/SG2 shared repeat density is plotted, not the SG2/SG3 shared repeat density.

Thank you, we have corrected this error which was due to a change in subgenome names to match Chaudhary 2020. Since we show that the original subgenome

partitioning proposed by Kagale et al. is wrong, we have rewritten the text to avoid directly referring to that older and incorrect notation.

- Supplementary note 2 : spell out 'EDF3' as Extended Data Figure 3.

Thanks. Done.

- Figure 5 : the labeling of V-mers, I-mers... below the x-axis in panel a is counterintuitive, these labels should be on the y-axis. I think it is also worth explaining in the text why the 'V-mers' for instance are better able to distinguish the *F. vesca*-related subgenome of the octoploid than the *F. vesca* genome itself (differences in color intensity on Figure 5b). I'm assuming this is due to divergent transposon activity since their last common ancestor ?

Thanks. We have revised the figure and noted clearly throughout that in these heatmaps columns represent chromosomes and rows are 13-mers.

Regarding the V-mers, as noted the 13-mers that differentiate subgenomes are computed using only the octoploid sequence. The fact that *F. vesca* chromosomes also share one set of k-mers is evidence for a relationship between *F. vesca* and the V-subgenome. Your assumption is correct: since the V-enriched k-mers are defined relative to the octoploid, they include repetitive activity in the V-progenitor after it diverged from the lineage leading to extant *F. vesca*. (Similar considerations apply to the I-subgenome vs. *F. iinumae*.)

- Figure 5 : reference in panel e legend to supplementary note 3 should be to supplementary note 5.

Thank you, we have addressed this.

- line 316 : spell out '(EDF. 5g)'.
- Extended data figure 5 legend erroneously refers to Figure 1a.

Thank you, corrected.

- supplementary note 3, 4 and 5 : figure and note references in these sections are incorrect. A lot of the text in these sections is redundant with the main text.

Done

- supplementary note 5 : Last two sentences of this supplementary note need to be reformulated (and references added).

Thanks. This has been corrected (now Supplementary Note 7).

- line 323 : reference to a non-existing Figure 1e.
- line 336 : 'share ancestry' should be 'shared ancestry'.
- several of the references contain formatting errors.

Thanks, these three items have been addressed.

Reviewer #3 (Remarks to the Author):

In the paper entitled “Transposon signatures of allopolyploid subgenome evolution”, the authors present a method to identify chromosome subgenomes in allopolyploid species. This approach is novel (though I do not follow the latest publications in the field) and can contribute to the analysis of subgenomes and their diploidization within allopolyploid animal and plant genomes. I question whether three examples are sufficiently demonstrating the robustness or applicability of the proposed approach. What is really needed is to spend some more time on explaining why these model groups were analyzed (not other). I failed to find any note on the applicability of the developed approach in old(er) polyploids, that is considerably diploidized genomes, such as “diploid” *Brassica* spp. (*B. rapa*). This should be discussed in the Discussion section and of course it would be great to include analysis of at least one of more diploidized genomes.

Better explanation of models. Thanks, we should have provided more motivation for these examples, and have added it in the revision. Briefly we chose the three examples (tetraploid goldfish/carp, hexaploid *Camelina sativa*, and octoploid strawberry) to demonstrate the applicability of our method to different ploidy levels and, for higher polyploids, address the order of hybridizations. In each case, there is some hypothesis to confirm or question to address.

For carp/goldfish, there is the question of whether the tetraploidy in these two related species is shared. This is now a relatively well-settled question based on comparisons with diploid ($2n=50$) relatives, and we show that our method (1) recapitulates the known subgenome structure obtained by previous work, providing confidence in our method, and (2) shows based on shared k-mer markers that the carp and goldfish descend from the same tetraploidization event (or at least share the same two diploid progenitor lineages). Our method also draws attention to goldfish GF33, which was noted by the original authors as anomalous based on gene content and protein-coding gene phylogeny.

For hexaploid *C. sativa*, two different subgenome hypotheses have been put forward: the original proposed by Kagale et al. 2014 based on the *C. sativa* genome sequence alone, and a different and definitive subgenome partitioning was proposed by Chaudhary et al. 2020 using extensive additional sequencing data from related diploids and tetraploids. We show that with our method the correct answer could have been obtained from the 2014 *C. sativa* sequence alone. (In fact, we had this result before seeing the 2020 Chaudhary et al. paper!) So again this analysis provides validation of our method. Furthermore, we find additional differentiation between the subgenomes of the tetraploid progenitor, information which is not available by comparison with extant diploids or the tetraploid *C. macrocarpa*.

Finally, we discuss the complex case of the octoploid strawberry, which has a long and contentious history. Multiple different subgenome hypotheses have been proposed. We show that our approach clearly defines four sub-genomes with strong statistical support. Surprisingly our four-subgenome partitioning of octoploid strawberry is different from previously proposed four-sub-genome partitions, but is consistent with all previous analyses (most of which fail to differentiate two of the four subgenomes, as described in text.)

Additional examples including other tetraploids. As suggested, we have added a few new examples as validations of our method and to help define its limits. These include:

- Two related allotetraploid cottons (*Gossypium hirsutum* and *G. barbadense*). Our method uses only intrinsic features of the polyploid genome, without any external data inputs from related diploids, to recover the well-established features of these two species, including their shared allotetraploidy. This parallels the shared allotetraploidy in carp and goldfish discussed in the original submission.
- Allotetraploid tobacco (*N. tabacum*). Again without reference to any external data, we recover the known subgenomes of tetraploid tobacco as well as the known inter-sub-genome rearrangements. This analysis uses a Hidden Markov Model for subgenome assignment and rearrangement analysis that is now added to the code base.
- Allotetraploid *Arabidopsis suecica*. *A. suecica* is a recent allotetraploid whose analysis was specifically requested. As expected, our approach recapitulates the known *A. thaliana* and *A. arenosa* subgenomes of *A. suecica*, again without any knowledge of the *A. thaliana* or *A. arenosa* diploid genome sequences. Although we find k-mers that separate the two sub-genomes, they are not (individually) statistically significant. This suggests that treating each k-mer independently may lead to overly conservative p-values.
- “Mesopolyploid” *Brassica rapa*. The “mesopolyploid” *B. rapa* arose by a ancient triplication (estimated to have occurred ~10 mya) from a diploid crucifer ancestor. (Cheng, Feng, et al. "The common ancestral genome of the *Brassica* species." The *Brassica rapa* Genome. Springer, Berlin, Heidelberg, 2015. 97-105.) Due to extensive rearrangement since the hexaploidy, contemporary *B. rapa* has only n=10 chromosomes. These have been partitioned into homoeologous chromosomal segments and assigned to putative subgenomes based on fractionation level as LF, MF1, and MF2 subgenomes (for least, medium, and most-fractionated, where fractionation is synonymous with gene loss). As now noted, our method did not find any consistent repetitive k-mer signal separating these putative subgenomes. We have discussed the analysis of *B. rapa* in a newly added subsection “Limitations of the method.” Since our method relies on detecting sequences relicts of ancient transposon activity, we anticipate that paleopolyploids that formed more than ~five million years ago would have reduced sub-genome-specific repetitive signals due to accumulated mutations and/or transposon turnover (deletion of old TE-derived sequences). Thus it is not unexpected that subgenomes of very old polyploids cannot be easily differentiated based on repetitive content.

As requested by the reviewers we also now provide the codebase in a GitHub repository (<https://github.com/amsession/Kmer-based-Subgenome-Mapping>), with a tutorial on how to run the code, using allotetraploid *Brassica napus* as a test dataset. These results are included as Extended Data Figure 1.

Introduction

It is incorrectly stated that the only pathway how allopolyploids are formed is hybridization followed by genome doubling. As the authors know an allopolyploid can be formed also in a single step or in a single step with preceding chromosome-set doubling in parental species. P2, L56: Please correct and rephrase: “... to be particularly controversial in part due to...”.

Thanks. We have changed “interspecific hybridization followed by genome doubling” to “interspecific hybridization and associated genome doubling” to allow for fusion of unreduced gametes, diploid gametes from autotetraploids, triploid bridges and other

phenomena that (eventually) lead to tetraploid AABB populations from the hybridization of AA and BB diploids. While the details of these processes are important for the initial origin of allopolyploids we prefer not go into them in detail, since they are not relevant to our analyses that are focused on the genomes of established allopolyploid species.

P3, L85: Figure 1. Please consider to clarify in the caption what the orange and blue flashes indicate. Fig. 1c fails to show what is described in its legend (“progenitors accumulate unique complements of transposable elements”).

Thanks, we have clarified this as suggested, and extensively reworked the Figure 1 legend.

P3, L105: “An important contrast is that since we do not typically have a training set of chromosomes of known provenance, however, without a training set of known authorship, we must bootstrap the identification of discriminatory DNA “words” from chromosome comparisons.” Please rephrase this sentence as it is rather unclear and somewhat interrupts the otherwise quite clear narrative of this paragraph.

We have attempted to clarify this in the revised text lines 114-116.

P3, L108: How k-mers were specific to transposable elements? Were genes masked on pseudo-chromosomes to identify only k-mer markers for TEs? Not specified in text here neither in Methods.

K-mers were computed on the unmasked chromosomes as now described in Online Methods and Supplementary Notes. We then find that k-mers with the relevant subgenome-specific pattern (high copy on a subset of chromosomes, one per homoeologous group) tend to overlap with transposable elements. (The use of $k=13$ means that they are typically not microsatellites, and as multiple figures now show they are distributed across chromosomes and are not concentrated as specifically sub-telomeric or centromeric repetitive elements. We have tried to clarify this in the text including more pointers to the Supplement.)

The specific connection between sub-genome-enriched k-mers and transposable elements is made by mapping the sub-genome-enriched k-mer to a genome and comparing these k-mer footprints with annotated repetitive regions. We do this for strawberry and *Camelina*. A strength of our method, however, is that we do not need to have such a TE annotation to apply the method and identify sub-genomes. We specifically focus on a dedicated transposable element annotator like LTR-Harvest for this since RepeatMasker is a distance-based algorithm that does not always identify both long-terminal repeats as part of the same element.

P4, L117-119: Here or elsewhere the authors should justify the choice of their allopolyploid genomes. Why not *Arabidopsis suecica* or *Tragopogon* spp. (for example)?

Thanks, we should have provided more motivation for these examples, summarized at top of the response, and have attempted to make this clearer in the revised text. Briefly, we chose the three examples (tetraploid goldfish/carp, hexaploid *Camelina sativa*, and octoploid strawberry) to demonstrate the applicability of our method to different ploidy levels and, for higher polyploids, address the order of hybridizations. In each case, there

is some hypothesis to confirm or question to address. Our goal was to not to provide a comprehensive analysis of recent polyploids, but rather to demonstrate that our method is broadly applicable and tackle several cases where open questions remain.

As requested we have now added a few more case studies where ground truths are available including two species of allotetraploid cotton and allotetraploid tobacco. In these cases the “answer” is already known because the diploid progenitors (or their close relatives) are extant, so they serve mostly as additional validation of our method, and to demonstrate that we can differentiate sub-genomes based on intrinsic features of a polyploid genome without reference to external information.

As suggested, we have now also analyzed *A. suecica* (Supplementary Note 10, Extended Data Figure 9). Here the method clearly differentiates the *A. thaliana* vs. *A. arenosa* subgenomes without reference to those diploids. Our statistical approach, however, tests each k-mer independently, so the p-values are conservative. We now note this in a newly added “Limitation of the method” section. We could not analyze *Tragopogon* since we could not find a suitable published genome sequence, but would be happy to analyze it!

B. napus is now provided as a worked example in the newly added GitHub, which contains the basic perl and R scripts for usage on tetraploids, hexaploids, and octoploids. (<https://github.com/amsession/Kmer-based-Subgenome-Mapping>), and a tutorial showing how to use the method with tobacco and cotton. These results are included as Extended Data Figure 1.

P4, L120: I am afraid that this entire sentence is messy and even repeated reading does not help to get some clear message. My understanding is that the authors confirm the existence of a tetraploid ancestral genome of *C. sativa*, but it is less clear what is meant by “missing diploid ancestor”. This diploid ancestor was identified in the recently published (and cited) Plant Cell paper. Btw. what is SG217? I see now, that means SG2. By comparing the two cited papers, the Plant Cell paper gives much clearer answer on the parentage of the *C. sativa* genome than the G3 paper. This should be taken into account.

Thanks. We have streamlined this discussion to emphasize that our approach partitions hexaploid *C. sativa* into three distinct subgenomes, two of which are closely related despite differing chromosome numbers. (See lines 138-146 of revised manuscript.) This analysis is conducted using only the hexaploid *C. sativa* genome sequence, but matches the results found by comparing the genome sequences of hexaploid *C. sativa* with diploid *C. hispida* and tetraploid *C. macrocarpa*. Thus independent evidence confirms our findings providing further confidence in the method. In addition, we find 13-mers that identifies the chromosomes derived from the tetraploid parent (denoted N6N7 by Mandakova) and shows that this *C. macrocarpa*-like parent is itself an allopolyploid despite being derived from the diploid *C. neglecta* (N7). We agree that the Mandakova et al. 2019 Plant Cell paper gives a very clear cytogenetic account of the origin of hexaploid *C. sativa* based on fluorescent *in situ* hybridization. Unfortunately, however, the chromosome numbering used in that paper is not obviously related to the chromosome labels of the sequenced *C. sativa* genome (e.g., Kagale et al. 2014, Nature Comm.). Thus we cannot directly associate Mandakova et al's chromosomes to sequence IDs. In contrast, the Chaudhary et al. G3 paper works directly with the sequenced genome. We have tried to clarify this in the now streamlined main text.

P4, L125: This paragraph should be re-written to set the scene properly. Using “controversial”, “other observations” and “novel partitioning” without giving corresponding details makes this paragraph more or less useless. I was not able to conclude anything from this text. If the order how parental genomes “were added” into the 8x genome was inferred, did you also identify the (other) two subgenomes discussed? The last sentence is totally enigmatic (what is “neutral explanation”? *F. vesca* subgenome if *F. vesca* subgenomes are introduced above? What observations from allohexaploid *Camelina*?

Thanks we have attempted to clarify this and set the (complex) scene in lines 147 and following, as well as in the strawberry results section. Briefly:

- It was previously known that octoploid strawberry contains an *F. vesca*-like “V” subgenome; an *F. iiumae*-like “I” subgenome, and two additional subgenomes (a total of 14 chromosomes) whose identity is controversial. There are two senses in which it is controversial. First, how are these 14 chromosomes partitioned into two subgenomes of 7 chromosomes each? There are several conflicting published “subgenome assignments” based on various considerations, without consensus. Second, what extant species are most closely related to these two subgenomes, and therefore are plausible source lineages? Since there may be extinct diploid or other ploidy strawberry lineages, there may be no close extant relatives to these lineages. The suggestion by Edger et al. 2019 that these non-V, non-I subgenomes are related to *F. nipponica* and *F. viridis* is a claim about both of these questions, which contributes to the confusion. Using our method, we identified four subgenomes of octoploid strawberry, each with distinctive ancestry as determined by 13-mer/transposable element content. Consistent with prior well-known findings, two of these are the “V” and “I” subgenomes.
 - By our 13-mer method, we can partition the remaining 14 chromosomes into two subgenomes of $x=7$ chromosomes, which we call T1 and T2. These (1) do not agree with Edger et al., or other published proposals, and (2) do not have any clear relationship to other extant diploid strawberries based on the published literatures. Thus while we identify the stable subgenomes T1 and T2 within octoploid strawberry, we do not know the progenitor lineages themselves, which may be extinct.
 - We refer to these non-V, non-I subgenomes as T1 and T2 since 13-mers shared by T1 and T2 but absent from both V and I suggest that T1 and T2 were united into a tetraploid prior to incorporation into the octoploid lineage. Taking into account the timing of this shared activity (using LTR divergence to determine time-since-insertion) we hypothesize the specific series of speciation and allo-hybridization events that led to octoploid strawberry (Figure 5h). This defines the “order” in which subgenomes are added to eventually produce the octoploid.

Regarding the enigmatic “neutral explanation,” this was unnecessarily cryptic, and we have expanded on it in the main text. According to the simplest model for gene duplication (in the style of Force et al.), homoeologous genes are initially redundant, so that (to a first approximation) mutations that delete or inactivate one copy will be neutral, or nearly neutral, and can drift to fixation, leading to gene loss. Similarly, many regulatory mutations that reduce gene function (by diminished expression, reduced catalytic effect, etc.) may also be expected to be neutral or nearly so. It follows that evolved polyploids are expected to have reduced gene content and generally weaker gene function (e.g., expression) than related diploids.

Now consider the formation of a higher polyploid, in which a diploid hybridizes with an extant polyploid that has already undergone gene loss/function reduction. The subgenome contributed by the diploid enters the new higher polyploid at full strength, while the other subgenomes that have already been evolving in a redundant context are lost or of reduced function. The added diploid subgenome will be seen as “dominant” to the others (based on reduced gene loss and more robust gene expression). But this is simply a consequence of the order of addition.

Methods

Origin of data not presented, which sequences/genome assembly versions were used? What was the source of transposable element sequences (and their distribution across the genomes)? Please add the information. Nothing is said about how the k-mer length was chosen. (Were other k-mer lengths tested?)

Supplementary Table 1 includes the specific genome sequences/versions used for the analysis. For strawberry and *C. sativa*, we ran LTR-Harvest ourselves on the genome fasta file using default parameters. We include the LTR-Retrotransposon families now in Supplemental Data Files 6-7.

We now note in the text that we typically use $k=13$, as done for all analyses in the paper, corresponding to k-mers that occur by chance once per ~33 Mb. We typically use odd values of k to avoid double counting palindromes. The current version of the analysis counts reverse-complemented k-mers as identical, removing the need for odd k lengths, however the odd-k-mer-length preference has stuck. In general short k-mers are preferred for computational speed and to ensure a high copy number. We generally do not use k less than 11 to avoid microsatellites, for which there is no plausible biological mechanism for sub-genome-specific accumulation. In some cases, longer k may be useful. The specific value of k may need to be tuned to identify marker sequences that are specific to one sub-genome vs. another, which in turn depends on substitution rate and the timing of speciation and allo-polyploidization. This is now noted but must be addressed on a case-by-case basis.

P5, L170: What tool was used for clustering

P5, L173: What method did you choose? Hierarchical clustering or other methods?

We used hierarchical clustering, implemented in R, using a distance function defined by $1-r$, where r is the Pearson correlation coefficient between the vectors of chromosome counts for two k-mers, as now noted more clearly in the text and Online Methods. This was initially discussed briefly in Supplementary Notes 1, 2, 3 but we agree that it must be noted in the main text. We expect that other clustering methods in k-mer space will produce the same results, and have also used k-means clustering in some cases, but have not systematically tested other clusterings.

P5, L188: No mention whether ANOVA assumptions were tested before using ANOVA.

Thanks for this question, now addressed in lines 224-233.. ANOVA assumptions include (1) normal distributions of (log-transformed) k-mer density per base-pair across k-mers; (2) equal variances of the two sub-genomes (levels or factors in the ANOVA language); and (3) independence. We have now noted these assumptions in text, and added figures

showing the relevant log-normal distributions as Extended Data Figure 10. In addition, we also applied the non-parametric Dunn's test, which produces results that generally agrees with the results of Tukey's test, albeit with fewer significant 13-mers.

For large datasets, standard tests for normality (e.g., Shapiro-Wilks or Kolmogorov-Smirnov tests) are generally viewed as too conservative for large data sets, and we rely on visual inspection. The distributions of 13-mer count/bp per subgenome appear bell-shaped (see new Extended Data Figure 10). The distributions are also normal when viewed at the chromosome level with the exception of Goldfish33, which was noted as misassembled by the authors of the original paper (data not shown; Supplemental Note 3). Similarly, within each species the histograms of k-mer density are very similar and the "equal variances" assumption is a reasonable approximation. ANOVA is known to be robust to minor variations in normality (see, e.g., <https://online.stat.psu.edu/stat500/lesson/10/10.2/10.2.1>) Finally, chromosomes are independently assembled, and 13-mers count per chromosome is measured independently, satisfying the requirement of independent measurement.

Results

p. 5: I understand that that it is some sort of jargon (practical usage), however, calling genomes with 50 chromosome pairs as paleotraploids and those with 25 pairs as diploids seems to be funny, particularly in the paper focused on this phenomenon. Consider some short explanation or rewording. I think this entire paragraph has to be rewritten to ease its understanding. For example, I was not able to comprehend where from the A subgenome comes from, what are L and S subgenomes? As there is no useful figure accompanying this text, its content cannot be understood.

We have attempted to clarify this in lines 275-291. The ancestral cyprinid state is $2n=50$. Thus barbs, or barbels, (*Barbus* sp.) are genetic and cytogenetic diploids typically with $n=25$ pairs of chromosomes.

Allotetraploid carp and goldfish, however, have $2n=100$ chromosomes per somatic cell, i.e., $n=50$ pairs of chromosomes. This is twice as many as the ancestral state. Both allotetraploid are genetic diploids in that they have $n=50$ genetically-defined chromosome pairs that consistently pair and recombine during meiosis. No meiotic pairing or recombination between homoeologs has been described. The 50 pairs of chromosomes in these allotetraploid species can be partitioned into two subgenomes, each a sets of 25 chromosome pairs.

We use "paleotetraploid" to indicate that the origin of these allotetraploids is ancient, rather than due to a recent hybridization event (sometimes, with more jargon (that we have tried to avoid!) called neo-tetraploids).

In the revised text we have tried to make clear that A/B, P/M, and S/L are the names assigned to the subgenomes of carp and goldfish by various authors. According to our 13-mer analysis, A, P, and S are synonyms for one of the subgenomes, and B, M, and L are synonyms for the other.

L225: " $2n=40$, $p=3$, $x\sim 7$, including one chromosome fusion)" I was not sure what is the meaning of "one chromosome fusion"?

A hexaploid formed from an ancestral $x=7$ karyotype would have $n = 3x = 21$ chromosome pairs in a diploid somatic cell. *C. sativa*, however, has $n=20$ chromosome pairs, which is accounted for (in our model, and also in Kagale 2014, Mandakova 2019, Chaudhary 2020) by one chromosome fusion. So instead of $3 \times 7 = 21$ chromosome pairs there are $20 = 7 + 7 + 6$ chromosomes. This is now explicitly discussed in the text lines 314-317.

I. 249 onwards: After checking the two cited papers, I conclude that the authors are intentionally bagatelizing these published results and conclusions, and over-emphasize their methods and results.

It was not our intention to minimize the importance of these prior papers, and have rewritten this section to make it less obscure (see lines 314-350). Originally we had given a chronological presentation that emphasized the difference between the initial *C. sativa* genome paper Kagale et al 2014, which used an *ad hoc* method to partition the genomes into sub-genomes 1, 2, and 3 based on comparisons to a distant outgroup, and the more recent Chaudhary et al. 2020 paper, which uses data from other related diploids to propose a different sub-genome hypothesis. This second proposal is presumably the correct answer, because it draws on additional sources of information that were not available to Kagale et al.

To demonstrate the power of our method, we independently reach the same conclusion as Chaudhary et al. using only the information contained in the hexaploid genome. In this way we (1) independently confirm Chaudhary's finding, and (2) show that the same conclusion could have been drawn five years earlier using only the hexaploid without reference to any other data. We also report a new finding that extends the results of Chaudhary et al. We find k-mers that are shared by the two *C. neglecta*-like chromosomes (N^6 and N^7 in the language of Mandakova; together *C. microcarpa*-like in the language of Chaudhary), but also k-mers that differentiate them. This implies the N^6 and N^7 subgenomes of *C. sativa* were first united in an allotetraploid progenitor. Note that Mandakova et al. seem to number their chromosomes based on their cytogenetic analysis and provide no correspondence with the numbering of Kagale and Chaudhary et al. (Our analysis, however, provides this correspondence.)

I. 255: I was not sure what I should imagine under "order of hybridization of *C. sativa* subgenomes that is intrinsic to the hexaploid itself" The order of hybridization (among subgenomes) is intrinsic to the hexaploid (genome)? The statement that "the previous finding of the existence of a related tetraploid to two of the subgenomes of the hexaploid merely implies ancestry, while our analysis provides a strong positive signal for the order..." is simply false. Both papers, mainly Mandakova et al., report on the order how subgenomes were hybridizing. In fact GISH-based identification of subgenomes (based on repeats) is congruent with your inference (fig. 4c, d). As for figure 4, it is puzzling why the authors do not use chromosome IDs from the cited Plant Cell paper, as their graphic subgenome-chromosome arrangement would look more elegant than at present (fig. 4a).

Thanks, we apologize for being unclear.

Regarding "order of hybridization", we are saying that, using only information from the hexaploid genome, we conclude that (1) first N^6 - and N^7 -like progenitors hybridized to form an allotetraploid, and (2) later this allotetraploid hybridized with an H^7 -like diploid to

produce *C. sativa*. As we note, this has also been shown by direct comparison of the hexaploid with extant tetraploids and diploids, both cytogenetically (Mandakova) and through sequence comparison (Chaudhary). The main point we were trying to make is that the information about the order of hybridization is contained within the hexaploid sequence itself, and the same conclusion can be reached without reference to any external datasets. We have attempted to clarify this further in the main text.

Regarding chromosome ID's, we use the chromosome numbering associated with the genome sequence in Genbank and other repositories, provided by the Parkin group and used in papers describing genome sequence analyses (Kagale 2014; Chaudhary 2020). It is, for better or for worse, the standard chromosome nomenclature for *C. sativa*. In their cytogenetic work Mandakova et al appear to use a different chromosome labeling that has no apparent relation to the chromosome nomenclature in Genbank.

p. 8, L297: please reword this to improve the clarity „but this has been questioned by Liston et al.18; the proposal of extensive homoeologous exchange by Edger et al.46,47 would further weaken this signal.”

Thanks we have tried to clarify this further in the main text, lines 352 et seq.. Relative to the diploid progenitors, homoeologous exchange or replacement creates mosaic chromosomes moving segments (and the k-mers they carry) from one sub-genome to the other. If such exchanges were common, as proposed by Edger et al., this would tend to equalize the k-mer counts between sub-genomes. Since we find k-mers whose counts are differentially enriched between sub-genomes, we can conclude that homoeologous exchange, if it occurred, was limited to small fractions of a chromosome.

In order to show what homoeologous rearrangements would look like via our method, we have included a discussion of the well-studied *Nicotiana tabacum* (tobacco) genome which has had a number of post-duplication rearrangements that have been verified by mapping of diploid *Nicotiana* reads to the polyploid assembly (Supplemental Note 8; Extended Data Figure 8).

P9: We note that the accelerated rate of sequence change in octoploid relative to diploid strawberries, coupled with possible extinction of relevant diploids, suggest possible explanations for the failure of methods dependent on protein coding phylogeny to identify the T1 and T2 sub-genomes”. Perhaps I missed something but did you identify the two subgenomes? My understanding is that you identified two subgenomes (T1 and T2) being closer to I subgenome (than to V subgenome). Then, what is the difference to say “the failure of methods dependent on protein coding phylogeny”? What exactly is meant by “the failure”?

As noted above, using our methods we did identify the subgenomes T1 and T2, in the sense that we found statistically significant sets of k-mer markers that show that these two chromosome sets have evolutionarily distinct ancestries and must therefore have been contributed by different progenitors (Figure 5a,b,e). This is what is meant by subgenomes.

By contrast, approaches based on protein coding sequence phylogeny (i.e., comparing octoploid strawberry genes with those of diploid *Fragaria* species) have **not** recovered these T1 and T2 sets. For example, Edger et al. claimed that these same 14 chromosomes should be partitioned in a different manner, which was questioned by

Liston in their Matters Arising exchange Our statistical analysis of k-mers, however, clearly shows that the proposed subgenomes of Edger et al. (which they call 'nipponica' and 'viridis' based on supposed similarity of octoploid strawberry genes to diploids with these specific names) is **not** consistent with k-mer (i.e., repetitive sequence) patterns (Supplementary Figure 5, panel b). Feng et al. surveyed a more complete panel of diploid *Fragaria* species, and still failed to correctly partition the T1 and T2 subgenomes correctly. These approaches have therefore failed to properly identify the subgenomes of octoploid strawberry.

Regarding the relationship between the *F. iinumae* and the 14 chromosomes that comprise T1 and T2 we do find that in addition to repetitive k-mers that differentiate them, there are k-mers are shared by T1 and T2, and by I, T1, and T2. In this sense T1 and T2 are closer to *F. iinumae* than V is (Figure 5a). The (distant) relationship between the 14 chromosomes comprising T1 and T2 and *F. iinumae* is also supported by protein-coding analysis (Tenessen et al. 2014) and haplo-SNP analysis (Sargent et al. 2016). But, again, these papers do not separate these 14 chromosomes into T1 and T2.

P: 9, L:345. "Such signals presumably do not arise in *Camelina* and the cyprinids because the interspecific divergences are more clearly defined." I admit that this explanation is not clear to me. This argumentation is actually (to some extent) negating what was said about the two subgenomes in the hexaploid *C. sativa* genome – all evidence (here and published) points to close genomic and phylogenomic relatedness among the two subgenomes.

We apologize that this was not clear, and have attempted to clarify it in the revised text. Specifically regarding *C. sativa*, although two of the three sub-genomes of this hexaploid are closely related (derived from *C. neglecta*-like ancestors), we can still find repetitive k-mer markers that consistently differentiate them. This can be done from the hexaploid genome itself without the need for extant relatives of these two *C. neglecta*-like lineages.

p. 10, l.354: I do not understand the significance of this paragraph. This narrative hardly fits into Results. Although I value this argumentation of the authors, I was not able to realize their contribution to this matter. Did you document the dominance of SG3 or not? Similarly, I do not understand the meaning of the last sentence. Why do we need to know the order of hybridization events to define dominant subgenome(s)? My understanding, also based on herein proposed method, is that (sub)genome dominance should be inferred (in ideal case) blindly, without some a priori assumptions. Please clarify, modify the wording.

We have expanded this discussion further, see lines 479-511. As noted in the next response below, in a higher polyploid (i.e., beyond tetraploid) the order in which genomes are hybridized is expected to have a clear effect on subsequent genome evolution. We have expanded on this important point below and in the main text.

Regarding *Camelina* and SG3, it was already shown to be "dominant" based on gene retention by Kagale et al. (i.e., 'biased fractionation') and gene expression by Chaudhary et al.. We, and Chaudhary *et al.*, find that SG3 was the last-added subgenome (i.e., an SG3-like progenitor hybridized with an SG1-SG2-derived tetraploid to form hexaploid *C. sativa*). We are noting the correlation between SG3 as the most recently added and its "dominance" (as already noted by Chaudhary), and providing a

general explanation for this phenomenon (see below). We have expanded this discussion to make the connection to results clearer.

Also the next paragraph is confusing. I do not think that Mike Freeling, Jim Birchler and others somehow consider that genome (sub)dominance is independent of time and the order in which genomes merge in (allo)polyploids. It follows that I do not get how conclusions of Edger et al. differ from yours. I also do not fully understand what is meant by “neutral expectation”. What is “neutral” on expectation that redundant (duplicated) genes are more often lost or have modified expression? This paragraph is also more discussion than results.

Thanks for this comment. We agree that the originators of the “sub-genome dominance” concept may have more nuanced appreciation similar to ours. Here we are specifically referring to the conclusions of Edger et al. 2019 regarding strawberry, which echo other ideas in the literature. Edger et al. 2019 make a strong claim in their Discussion:

“Analysis of this [octoploid strawberry] genome allowed us to identify each of the diploid progenitor species, reconstruct the evolutionary history of the octoploid event, and investigate the evolution of a dominant subgenome. Our data support the hypothesis that subgenome dominance in an allopolyploid is established by TE-density differences near homeologous genes in each of the diploid progenitor genomes [Freeling 2012]. Furthermore, our results show that the *F. vesca* subgenome has increased in dominance over time by having retained significantly more ancestral genes and a greater number of tandemly duplicated genes than the other three subgenomes, and replaced large portions of the submissive subgenomes via homeologous exchanges.”

Citation: Freeling, M. et al. Fractionation mutagenesis and similar consequences of mechanisms removing dispensable or less-expressed DNA in plants. *Curr. Opin. Plant. Biol.* 15, 131–139 (2012).

The impression given throughout Edger 2019 is that *F. vesca* has acquired its “dominance” as a mechanism for “resolving epigenetic conflict” (as stated elsewhere in Edger et al. 2019). From our perspective, invoking an urge to “resolve conflict” is unnecessarily teleological. Similarly, as explicitly stated in this excerpt above, they claim “subgenome dominance in an allopolyploid is established by TE-density differences near homeologous genes in each of the diploid progenitors genomes,” citing the 2012 review by Freeling. This statement seems to suggest that the V-subgenome is dominant to other subgenomes by virtue of the TE-gene associations that were present in the diploid progenitor *F. vesca*. The claim that their results “show that the *F. vesca* subgenome has increased in dominance over time” suggests an ongoing process intrinsic to the *F. vesca* sub-genome, and reference homeologous exchanges that we do not find. Similarly, elsewhere in Edger 2019 the other subgenomes are referred to as “submissive”.

There is, however, a simple alternative model that does not rely on any specific “dominance” property of the *F. vesca* genome. This model is “neutral” in that it follows generally from the fixation of common mutations that reduce or eliminate redundant gene function as discussed by Force, et al., “Preservation of duplicate genes by complementary, degenerative mutations.” *Genetics* 151.4 (1999): 1531-1545. (PMID: 10101175) The key point is that polyploids evolve under conditions of genetic

redundancy (to the extent that homoeologous copies of genes are redundant), which leads to ongoing gene loss and diminished/sub-functionalized expression.

At the time of the final diploid-hexaploid hybridization event, the octoploid strawberry genome comprised a complete diploid *F. vesca* genome combined with the three non-V sub-genomes from the hexaploid progenitor. Notably, these three non-V sub-genomes had already evolved together for several million years as a hexaploid under conditions of genetic redundancy. Thus at the time the octoploid was formed, these non-V subgenomes were already partially degraded, i.e., “fractionated” by gene loss, and/or with diminished/sub-functionalized expression. In contrast the V-sub-genome, newly transferred from a functional diploid, was intact.

In this scenario, the V-subgenome is expected to have (1) a more complete gene complement and (2) more robust gene expression than the three non-V subgenomes. That is, it will appear to “dominate” them. But this “dominance” has nothing to do with any special feature of the *F. vesca* progenitor or its genome, other than that it was the last sub-genome added to the octoploid mix. This bias toward gene loss/reduction in gene expression in non-V sub-genomes is expected to persist over time, since the loss of a non-V-sub-genome gene (many of which are already expressed at lowered levels) is expected to have fewer negative consequences than the loss of a more highly expressed V gene. (Epistatic effects also likely contribute, and these also favor the V subgenome, since to the extent that non-V genes interfere with robust V function, they will be preferentially lost.)

Our point is that “subgenome dominance” in a hexaploid, octoploid, etc. can arise simply from the order of addition, and as a consequence of “neutral” processes akin to those discussed by Force, et al., 1999 rather than being intrinsically better suited to dominate, as implied by Edger et al. 2019. Although this mechanism cannot explain the asymmetric sub-genome evolution in allo-tetraploids (which are combinations of two diploid genomes that are both “intact” at the time of allo-tetraploidization), the “order of addition” phenomenon is a general feature of higher allo-polyploids, e.g., a diploid joined to a pre-existing hexaploid as in strawberry, or a diploid joined to a pre-existing tetraploid as in *Camelina*. Chaudhary et al. mention this in passing, and we have now also added a citation.

We have expanded the discussion in the main text to make these points more systematically.

Reviewer #4 (Remarks to the Author):

Review

This work by Session & Rokhsar identifies k-mers specific to sets of divergent chromosomes to distinguish divergent subgenomes in polyploids. This in silico version of molecular cytogenetics is an original way of looking at assemblies of complex genomes. Here, reanalysis of the conserved carp genome as well as dynamic plant genomes such as in *Camelina* and in the very complex *Fragaria* was used as a validation of the approach.

Not being the necessary expert of the “controversial” (sic?) *Fragaria* genome to ascertain to what extent outputs of the approach offers added-value, I can only report that results in that case as well as other case studies were convincing.

Thanks. Documentation of the controversy can be found in the Matters Arising exchange of Liston et al 2020 and Edger et al 2020, as well as Feng et al. 2021.

One of the main issue requesting clarification from my point of view is the extent to which tracked k-mers are derived from transposable elements. That the statistical toolbox used here can detect highly-repeated sub-genome markers is clear, but I have not read about the validation of k-mers being part of (retro?)transposons rather than e.g. tandem repeats (i.e. other than centromeric ones). On l. 319, it is simply stated that “13mers we define overlap with annotated families of retrotransposons”. This looks a critical validation for part of the results focused on transposons (e.g. dating). If a strict association (specificity) of tracked 13-mers with transposable elements is hardly apparent, I would suggest to skip most of related arguments (i.e. throughout).

Thanks. We have clarified this since it is an important part of our results (specifically, as noted, the dating of events in the history of the octoploid strawberry). Previously we have shown this to be the case in allotetraploids such as *X. laevis*, *B. hybridum*, *M. sinensis*, and *P. virgatum*. In addition, we have now added Supplemental Data Files 10-11 for strawberry and *Camelina* that include not only the full complement of retrotransposons we define using LTR-Harvest, but also the specific families that are marked by subgenome-enriched 13-mers for each subgenome.

We note that the association of sub-genome-specific k-mers with transposable elements provides the conceptual underpinning of our approach, since only transposable elements are able to spread specific sequence motifs between chromosomes. In particular, there is no process we know by which a specific tandemly repeated sequence could spread throughout a genome and in that way tag all chromosomes (in a progenitor species) with the same sequence.

In general, the presentation of the work could be improved. The core text reads well, although several lengthy descriptions and other typos in the spelling of figures and tables made it difficult to follow the details. Supplementary information is very rich and it is sometimes impossible to be confident about claimed justifications. I will not go beyond the effort of requesting a thorough check.

We apologize for these residual typos. We have reviewed the main text, figure legends, tables, and supplementary information to correct these errors and in general make sure that all claims are clearly justified to the reader.

An a more minor side:

I would refrain to relate allopolyploids to permanent heterozygosity. It is “fixed heterozygosity” of a very different kind that found in e.g. *Oenothera* that has been termed “permanent”. Two kinds of polyploids is at best schematic and poorly accounts for increasingly diverged progenitors. Similarly over-simplistic: genome doubling does not necessarily occurs “after” hybridization at the origin of allopolyploids, aso...

Thanks. We agree that “permanent heterozygosity” is a poor choice of phrase, since it could easily be confused with *Oenothera*-like behavior. We have removed this inessential comment and attempted to clarify the discussion.

Reviewers' Comments:

Reviewer #1:

Remarks to the Author:

The authors have satisfactorily addressed all of the comments I submitted in my original review. However, as documented by the quoted text inserted below, I subsequently (Nov 28, 2021) contacted Nature Communications by e-mail with a significant concern. Until this concern is satisfactorily addressed, I cannot recommend publication of this manuscript. It has not been addressed by the current revision.

My e-mail message:

"To Nature communications:

I had anticipated that I would have a second opportunity to review this manuscript, after the authors have responded to the initial reviews. If this is not the case, I would like to raise a new and serious concern, and strongly urge the editors to address it. Specifically, subsequent to my review submission, I encountered a paper that was published in 2019, and that is highly relevant to the manuscript under review, yet was not cited by the authors. This manuscript (Gordon et al., 2019, POLYCracker, BMC Genomics (<https://doi.org/10.1186/s12864-019-5828-5>) which I have attached, must have been known to at least one of the present authors due to their institutional affiliations. The current authors' neglect to mention this prior work, which describes a K-mer-based approach very much like that now reported seems to me to approach intellectual dishonesty. Accordingly, I strongly recommend that the manuscript by Session and Rokhsar should not be accepted for publication until they have adequately acknowledged this key prior work by Gordon et al.,, and commented on its relevance to their manuscript.

Reviewer #2:

Remarks to the Author:

The authors addressed my comments satisfactorily, and the additional case studies performed are a welcome addition to the manuscript, further clarifying the method's capabilities and limitations. I have no further comments except that the authors should double-check their text for typos. There is for instance an incomplete reference to Edger et al. on line 276 of the supplementary information, and on line 422 of the same document 'A. thaliana' should be 'A. arenosa'.

Reviewer #3:

Remarks to the Author:

The authors tried to resolve all comments and objections of the reviewers. I appreciate the high quality of author-reviewer debate. The revised manuscript version was improved as compared to the initially submitted version.

I have only two minor points to add. (1) the wrong form "macrocarpa" is used for *Camelina microcarpa*, and (2) the authors may want to refer to the recently published paper on the identification of the purported tetraploid *Camelina N6N7* genome (<https://onlinelibrary.wiley.com/doi/abs/10.1111/tpj.15931>). My understading is that the newly published paper and conclusions made by the authors are mutually congruent.

I am looking forward seeing the paper in print!

Reviewer #4:

Remarks to the Author:

This is the revised version of a manuscript that I reviewed a while ago for Nature Communication. Again, I find the topic exciting, and can see the potential for a useful tool. Given that I have no better hope than during last round about how my detailed comments would be treated in the context of this journal and provided the pressure to review quickly, I will again focus on a few issues with the hope that the authors want to publish solid and convincing work.

My major comment that k-mer were not shown to be derived from TEs was echoed by most reviewers has been addressed. I concur with the authors that mostly TEs can be expected, on theoretical ground, to show the targeted pattern. However, given claims here and the necessity to justify the approach, it seems a reasonable request to have evidence supporting that 13-mers are properly associated to TEs diverging among (progenitors of) subgenomes.

I am not yet convinced by evidence that came under my eyes. Messy SI files with improper labelling (apparently a recurrent trick of that manuscript system) and poor description left me uncertain to have inspected the appropriate Supp Files 10 and 11. I finally scrutinised all SI in great details and, apart from typos and awkward descriptions, I noticed nothing special; at least, no clear supporting evidence of the kind requested by several reviewers.

Furthermore, would evidence be crystal clear and fully supportive, it may be sufficient - although ambiguously partial - to report from 2-3 case studies only (as here attempted). As said above, it is not yet crystal clear and the whole approach thus appears unnecessarily questionable, leaving it in a state that is not as convincing as it could be. I can only recommend that the authors embark in a more systematic survey ideally linking their "trick" to the biology of the genome.

Although I spotted several small issues where arguments could be raised and incrementally revised for ever, I would like to only mention one aspect connected to my major comment above and that left me in some confusion. It was certainly a good idea to add new case studies offering pragmatic support and, despite sometimes superficial description, it mostly enabled to highlight possible caveats that the lack of theoretical background would have otherwise left hidden. More specifically, that the approach may not work properly anymore in older polyploids remains slightly unclear to me and would deserves some more detailed justification. Indeed, that TEs usually get increasingly fragmented and divergent with time is well known and logical, and this is very certainly a chief motivation for the use of short k-mers (again, something for which a justification has been requested and that is here superficially addressed). However, it remains unclear to me how the authors connect the TE life-cycle to their approach, results and conclusions. Here, the proverbial "4.5 MY half-life" of TEs is simply referenced from a single paper in grasses (although no grass was included in this study) to justify that 6-7 MY old polyploids such as Brassica do not work properly... This is at best sub-optimal, possibly misleading. Some more in depth justification would be desirable. I guess that it would be fair to say that not much is known about the rate of TE decay. Accordingly, it would be rather useful to discuss how long it may take for TE sequences to diverge to such an extent that sufficiently repeated 13-mer motifs cannot be identified anymore; for instance relying on studies along the line of Maumus & Quesneville 2014 (Ancestral repeats have shaped epigenome and genome composition for millions of years in *Arabidopsis thaliana*. Nat Commun 5:4104. <https://doi.org/10.1038/ncomms5104>) or simple probabilities.

Reviewer #1 (Remarks to the Author):

The authors have satisfactorily addressed all of the comments I submitted in my original review. However, as documented by the quoted text inserted below, I subsequently (Nov 28, 2021) contacted Nature Communications by e-mail with a significant concern. Until this concern is satisfactorily addressed, I cannot recommend publication of this manuscript. It has not been addressed by the current revision.

My e-mail message:

"To Nature communications:

I had anticipated that I would have a second opportunity to review this manuscript, after the authors have responded to the initial reviews. If this is not the case, I would like to raise a new and serious concern, and strongly urge the editors to address it. Specifically, subsequent to my review submission, I encountered a paper that was published in 2019, and that is highly relevant to the manuscript under review, yet was not cited by the authors. This manuscript (Gordon et al., 2019, POLYCracker, BMC Genomics (<https://doi.org/10.1186/s12864-019-5828-5>) which I have attached, must have been known to at least one of the present authors due to their institutional affiliations. The current authors' neglect to mention this prior work, which describes a K-mer-based approach very much like that now reported seems to me to approach intellectual dishonesty. Accordingly, I strongly recommend that the manuscript by Session and Rokhsar should not be accepted for publication until they have adequately acknowledged this key prior work by Gordon et al., and commented on its relevance to their manuscript.

Thanks for reviewing our manuscript a second time. Please note that we had not seen your email about POLYCracker until just now, when we received it with this most recent round of reviews. We take concerns about intellectual honesty very seriously. Below we document our priority in using k-mers to differentiate sub-genomes based on a series of public presentations made by us.

As emphasized throughout our manuscript, Session et al. 2016 is the first use of differentially distributed transposable elements to characterize subgenomes in a polyploid (*Xenopus laevis*). We had also clearly presented our ideas on k-mer analysis in both public meetings (e.g., PAG 2017 and 2018, see below), in earlier annual national *Xenopus* meetings, and in internal meetings with our JGI colleagues. The application of our k-mer approach was also described by us in other papers including their application to other genomes, now subsequently published and cited in Mitros, T., Session, A.M., James, B.T. *et al.* "Genome biology of the paleotetraploid perennial biomass crop *Miscanthus*." *Nat Commun* 11, 5442 (2020). <https://doi.org/10.1038/s41467-020-18923-6> and Lovell, J.T., MacQueen, A.H., Mamidi, S. *et al.* "Genomic mechanisms of climate adaptation in polyploid bioenergy switchgrass." *Nature* 590, 438–444 (2021).

<https://doi.org/10.1038/s41586-020-03127-1>. The present paper is intended to provide a statistical framework for these analysis and an extension to higher polyploids.

We were, to put it mildly, surprised that our JGI colleagues published their 2019 paper without citing our prior work (e.g., Session et al. Nature 2016) or their prior discussion with us. Our prior and contemporaneously ongoing work, and our discussions with Levy, Gordon, and Vogel about the method were, inexplicably, not cited or acknowledged in the “POLYcracker” paper. Our response was to simply to ignore the POLYcracker publication, which we agree is not good scientific practice.

In the present revised manuscript we now acknowledge the existence of the POLYCracker paper by citing at several point in the manuscript as an alternative k-mer implementation, but also include a citation to our 2018 PAG Abstract (copied below) that lays out the k-mer based approach well in advance of POLYCracker. (For reasons that are not clear to us, the 2017 PAG abstract is not available on the PAG website, and so is not cited, but is provided below from our email records.)

2018 PAG presentation (poster):
Plant and Animal Genomes (PAG) XXVI January 13-17, 2018.
Adam Session
P1250 Genome Evolution following Polyploidy in Grasses.
Abstract (retrieved from https://plan.core-apps.com/pag_2018/abstract/70f25ba3db39e46859cce4932bf6e247)

“Polyploid organisms can arise due to genome doubling without cell division (autopolyploidy) or doubling after interspecific hybridization (allopolyploidy), resulting in more than two chromosome sets per somatic nucleus. Similarity between such homeologous chromosomes presents a barrier to accurate polyploid genome assembly. In recent years, better assembly methods have overcome this hurdle, enabling quantitative exploration of hypotheses concerning the mechanism of individual polyploid formation, as well as the initial response of genomes to polyploidy.

We hypothesized that one positive marker of allopolyploidy would be transposable elements that differentially expanded in the two diploid progenitors prior to hybridization. To find such markers, we developed a method of using k-mers (identified by Jellyfish) to identify commonly occurring “words” that distinguish homeologous chromosomes. In the polyploid genomes where we found this positive signal, the words that split pairs of homeologous chromosomes overlap, allowing the identification of coherent subgenomes that are each descended from each diploid progenitor. When mapped to the genome, these differentiating 15mers align to the long-terminal repeats(LTRs) of retrotransposons. Studying the sequences divergence of these LTRs we can estimate

the timing of the hybridization of polyploidy. In contrast, the divergence of progenitor species typically predates polyploidization.

We show that two tetraploid grasses, *Panicum virgatum* and *Brachypodium hybridum*, are allopolyploid by this method. The test produces a null result in some tetraploids, such as the grass *Miscanthus sinensis*. We also investigated the evolution of protein-coding genes following duplication, hoping to use the repeated natural experiment of polyploidy to model gene decay following redundancy.”

This poster is a large file but we have available for anonymous download from Google Drive

<https://drive.google.com/file/d/12iLWXnidVRlwotUmvkJh2R6tU4xR6H17/view?usp=sharing>

2017 PAG presentation (invited talk)

Plant and Animal Genomes (PAG) XXV January 14, 2017

Adam Session

W313 Repetitive Elements Reveal the Evolutionary History of Polyploid Grass Genomes Workshop: 4207 Evolution of Genome Size

Talk title retrieved from <https://pag.confex.com/pag/xxv/meetingapp.cgi/Paper/24986>

While the abstract is not available on the PAG archive site, we attach below a copy of the email sent to PAG including the abstract:

from: pag@schicago.com via lbl.gov
reply-to: pag@schicago.com
to: amsession@lbl.gov
date: Nov 18, 2016, 10:20 AM
subject: PAG XXV - Abstract Acceptance Email
mailed-by: lbl.gov

Dear Adam Session,

This email confirms your abstract, Repetitive Elements Reveal the Evolutionary History of Polyploid Grass Genomes, has been accepted for presentation at the Plant and Animal Genome XXV Conference, January 14-18, 2017 at the Town & Country Hotel & Conference Center in San Diego.

Polyploid organisms can arise due to genome doubling without cell division (autopolyploidy) or doubling after interspecific hybridization (allopolyploidy), resulting in more than two chromosome sets per somatic nucleus. Similarity between such homeologous chromosomes presents a barrier to accurate polyploid genome assembly. In recent years, better assembly methods have overcome this hurdle, enabling quantitative exploration of hypotheses concerning the mechanism of individual

polyploid formation (allo vs auto), as well as the initial response of genomes to Polyploidy.

We hypothesized that one positive marker of allopolyploidy would be transposable elements that differentially expanded in the two diploid progenitors prior to hybridization. To find such markers, we developed a method of using k-mers (specifically 15-mers identified by Jellyfish) to identify commonly occurring words that distinguish homeologous chromosomes. In the polyploid genomes where we found this positive signal, the words that split pairs of homeologous chromosomes overlap, allowing the identification of coherent subgenomes that are each descended from each diploid progenitor. When mapped to the genome, these differentiating 15mers align to the long-terminal repeats (LTRs) of retrotransposons in plants, and often align to DNA transposons in animals. By studying the sequences divergence of these LTRs we can estimate the timing of the hybridization of polyploidy. In contrast, the divergence of progenitor species typically predates polyploidization.

We use this method to show that multiple grass and vertebrate genomes are allopolyploid.

We also investigated the evolution of protein-coding gene expression following polyploidy, and its relationship with expanding transposable elements

Abstracts submitted for the poster session will receive their final poster number 3 weeks prior to the meeting.

If your abstract is for a workshop talk, contact your workshop organizer regarding the timing of your presentation or visit the PAG website at:

<https://pag.confex.com/pag/xxv/meetingapp.cgi>

If there is anything that needs addressing, please contact the abstract coordinator at:

Gerard Lazo - gerard.lazo@ars.usda.gov

We look forward to your attendance at PAG XXV 2017.

We also presented a similar poster to the SMBE 2018 meeting, whose abstract is linked here under POA-368: <https://evolgen.biol.se.tmu.ac.jp/SMBE2018/POA.pdf> (page 249 of abstract book; see also https://smbe.org/smbe/SMBE2018Meeting/mobile_app.html#abstract_pdf). The abstract is copied below:

Subgenome-Enriched Transposable Elements Reveal Allopolyploid Origins Following Whole-Genome Duplications

Adam M Session1 , Daniel S Rokhsar1, 2
1 Joint Genome Institute (United States),
2UC Berkeley (United States)

Polyploid organisms can arise due to genome doubling without cell division (autopolyploidy) or doubling after interspecific hybridization (allopolyploidy), resulting in more than two chromosome sets per somatic nucleus. Similarity between such homeologous chromosomes presents a barrier to accurate polyploid genome assembly. In recent years, better assembly methods have overcome this hurdle, enabling quantitative exploration of hypotheses concerning the mechanism of individual polyploid formation (allo vs auto), as well as the initial response of genomes to polyploidy.

We hypothesized that one positive marker of allopolyploidy would be transposable elements that differentially expanded in the two diploid progenitors prior to hybridization. To find such markers, we developed a method of using k-mers (specifically 15-mers identified by Jellyfish) to identify commonly occurring words that distinguish homeologous chromosomes. In the polyploid genomes where we found this positive signal, the words that split pairs of homeologous chromosomes overlap, allowing the identification of coherent subgenomes that are each descended from each diploid progenitor. When mapped to the genome, these differentiating 15mers align to the long-terminal repeats (LTRs) of retrotransposons in plants, and often align to DNA transposons in animals. By studying the sequence divergence of these LTRs we can estimate the timing of the hybridization of polyploidy. In contrast, the divergence of progenitor species typically predates polyploidization.

We use this method to show that multiple grass and vertebrate genomes are allopolyploid. We also investigated the evolution of protein-coding gene expression following polyploidy, and its relationship with expanding transposable elements.

Priority aside, our present manuscript has several key differences relative to the POLYCracker paper:

1. Our manuscript describes a well-founded statistical test for grouping chromosome-scale sequences into sub-genomes based on k-mer distribution. In contrast, the POLYCracker paper describes the spectral clustering of a large number of ~100 kb scaffolds but provides no statistical basis for this method. (We had previously applied our methods to sub-chromosomal scaffolds during the development of the *X. laevis* assembly, and presented this work at various public *Xenopus* meetings, prior to POLYCracker. But in the end this sub-chromosomal work did not find its way into the *X. laevis* publication and is not described in any publicly available document or abstract.)
2. In hexaploid and octoploid contexts (Camelina and strawberry, respectively) we describe shared signals between multiple sub-genomes and interpret them as

- implying shared ancestry among progenitors. While the POLYCracker paper applies its methods to hexaploid wheat, the aim is only to separate the sub-genomes and there is no mention of shared k-mer content between the A, B, or D subgenomes (any such signal would be hidden within the spectral clustering), or its possible interpretation.
3. We describe a Hidden Markov Model framework for identifying transitions in sub-genome state across a chromosome, as would be caused by homoeologous exchange and/or replacement. This allows such regions to be recognized in a standard statistically well-formulated framework. In POLYCracker, ~100 kb blocks are clustered without regard to chromosomal location. When projected back onto chromosomes these suggest transitions, but again without a statistical framework.
 4. We interpret our k-mers in terms of transposable element sub-families that were differentially expanded in polyploid progenitors. This allows us, for example, to infer the timing of key events in the formation of octoploid strawberry. In our most recent revision have expanded our discussion of the connection between k-mers and transposable element sub-families. In contrast, in the POLYCracker paper it was shown that while the same sub-genome assignments could be obtained using only repeat-masked regions, the relevant repetitive sequences were only characterized as “unknown” based on their repeatmasker annotation. (In our manuscript we now specifically describe common LTR retrotransposons in tobacco that are sub-genome-specific.)
 5. Most importantly, we use our sub-genome identification methodology to address an outstanding problem in polyploid ancestry (octoploid strawberry) and another problem in polyploid ancestry that was solved contemporaneously with our analysis (*Camelina*, whose ancestry we assessed based entirely from the hexaploid sequence; as we describe, our findings were confirmed by subsequent developments integrating data from multiple diploid and tetraploid *Camelina* spp.)

Reviewer #2 (Remarks to the Author):

The authors addressed my comments satisfactorily, and the additional case studies performed are a welcome addition to the manuscript, further clarifying the method's capabilities and limitations. I have no further comments except that the authors should double-check their text for typos. There is for instance an incomplete reference to Edger et al. on line 276 of the supplementary information, and on line 422 of the same document 'A. thaliana' should be 'A. arenosa'.

Thanks for reviewing our manuscript a second time and for your additional comments. We apologize for residual typographical errors and will review the manuscript again to be sure to catch any others.

Reviewer #3 (Remarks to the Author):

The authors tried to resolve all comments and objections of the reviewers. I appreciate the high quality of author-reviewer debate. The revised manuscript version was improved as compared to the initially submitted version.

I have only two minor points to add. (1) the wrong form "macrocarpa" is used for *Camelina microcarpa*, and (2) the authors may want to refer to the recently published paper on the identification of the purported tetraploid *Camelina* N6N7 genome (<https://onlinelibrary.wiley.com/doi/abs/10.1111/tpj.15931>). My understanding is that the newly published paper and conclusions made by the authors are mutually congruent.

I am looking forward seeing the paper in print!

Thanks for reviewing the manuscript a second time and for your helpful comments, which we agree have improved the manuscript substantially.

Thanks for catching the micro/macro typo! Its not clear how that error crept entered into the manuscript. We have checked that the correct "microcarpa" is used throughout. Thanks!

Thanks also for pointing out the very recent (August 2 2022!) paper by Mandakova and Lysak! Their findings provide further support for our assignment of chromosomes to subgenomes based solely on analysis of the polyploid genome itself. We will consult with the Editor whether it is more appropriate to cite this paper in the main body of the manuscript or as a "Note Added in Proof".

Reviewer #4 (Remarks to the Author):

This is the revised version of a manuscript that I reviewed a while ago for Nature Communication. Again, I find the topic exciting, and can see the potential for a useful tool. Given that I have no better hope than during last round about how my detailed comments would be treated in the context of this journal and provided the pressure to review quickly, I will again focus on a few issues with the hope that the authors want to publish solid and convincing work.

We appreciate your detailed comments, previously and now below, and have made good-faith efforts to address all of them within what we think is the scope of this manuscript. Please see below for further discussion.

My major comment that k-mer were not shown to be derived from TEs was echoed by most reviewers has been addressed. I concur with the authors that mostly TEs can be expected, on theoretical ground, to show the targeted pattern. However, given claims here and the necessity to justify the approach, it seems a reasonable request to have evidence supporting that 13-mers are properly associated to TEs diverging among (progenitors of) subgenomes.

I am not yet convinced by evidence that came under my eyes. Messy SI files with improper labelling (apparently a recurrent trick of that manuscript system) and poor description left me uncertain to have inspected the appropriate Supp Files 10 and 11. I finally scrutinised all SI in great details and, apart from typos and awkward descriptions, I noticed nothing special; at least, no clear supporting evidence of the kind requested by several reviewers.

We are trying! Its not clear to us what specific additional evidence you are seeking.

In **Supplementary Files 10 and 11** we provided complete data tables listing (1) all instances of LTR retrotransposons in the *Camelina sativa* and Strawberry genomes and a (2) list of those LTR retrotransposons that overlap sub-genome-enriched 13-mers. Since there is no canonical annotation of retrotransposons in *Camelina sativa*, we provided a complete list from our LTRHarvest analysis described here. We have now added **Supplementary File 12** including this information for tobacco.

While this is an overwhelming amount of information it is a complete documentation of the sequences we used for the analysis. In order to make this more clear to readers, we have added README files for these tarballs, explaining the content of each file, as well as the naming system we used for the retrotransposons.

We have also now added a new **Extended Data Figure 11** that shows phylogenetic trees of long terminal repeats of LTR retrotransposons, using tobacco as an example, colored by the sub-genome type of the corresponding chromosome. Subgenome type is based on the presence of sub-genome-specific k-mers as assigned by our method. These trees show clearly that (1) In large TE families, LTRs from from different sub-genomes form distinct clades, and (2) such sub-genome-specific TEs can

be recognized by our k-mer method (which does not require prior TE analysis and is independent of any specific TE type). While we show this analysis to connect our k-mer analysis to TEs, the k-mer method is substantially faster and simpler than a full TE annotation, and provides key sequence-specific markers that differentiate TE sub-families by their sub-genome type. The raw fasta alignments for tobacco, Camelina, and strawberry are included in the new **Supplementary Data File 13**.

Please note that the families and sub-families as assigned by tools like RepeatModeler/RepeatMasker may lump together sub-genome-specific TEs into a single large family, since its methods do not take polyploid genome organization into account. Additionally, RepeatModeler/RepeatMasker often do not take the larger retrotransposon structure into account, and may, for example, call the long-terminal repeats and inner sequences as different families of repeats. Thus it is not sufficient to simply ask if TE families or sub-families as defined by repeatmasker are differentially distributed along the genome, or analyze RepeatMasker sequences phylogenetically. We have added a brief discussion of this point.

Furthermore, would evidence be crystal clear and fully supportive, it may be sufficient - although ambiguously partial - to report from 2-3 case studies only (as here attempted). As said above, it is not yet crystal clear and the whole approach thus appears unnecessarily questionable, leaving it in a state that is not as convincing as it could be. I can only recommend that the authors embark in a more systematic survey ideally linking their "trick" to the biology of the genome.

It is not clear to us what a "more systematic survey" would look like.

1. We have shown that in many cases a polyploid genome can be partitioned into distinct subgenomes **without reference to external data from diploid relatives**, purely based on the chromosomal distribution of its intrinsic repetitive content. Repetitive here is used in its most general context, as sequences that occur repeatedly in one subgenome vs. another, which we identify using k-mers.
2. We analyzed **twelve** polyploid genomes in this paper alone, and cited publications of four other polyploids (tetraploid *Xenopus*, *Brachypodium*, *Miscanthus*, and switchgrass) where this method was previously applied.
3. Shown that when additional information about progenitors is available (e.g., goldfish/carp, partially for *Camelina* and strawberry), our partitioning of the chromosomes of the polyploid into subgenomes **using only the polyploid genome itself** is correct.

It is also unclear to us what you mean by "unnecessarily questionable." We have presented a statistically sound method and benchmarked it in multiple cases. We have tried to make our approach and results "crystal clear" and would welcome identification of specific points where further clarity could be achieved. We have shown that the phenomena is a general one, and noted limitations on the age of the allopolyploidy

events that can be studied with it, due to erosion of signal due to ongoing mutation. This is now clarified further (see also answer to your last question below).

We have also made very clear the connection between the “biology of the genome” and our method (or “trick”). Specifically, repetitive sequences arising from the activity of transposable elements in the progenitor(s) of an allopolyploid will mark descendant chromosomes. We show that as long as this repetitive signal persists (i.e., has not deteriorated by mutation), these can be tracked without comparisons to extant genomes related to these progenitors, which may be extinct or not sequenced.

The complement of transposable elements varies from genome-to-genome. Thus the specific nature of sub-genome-specific “repetitive elements” will of course vary across independent instances of allopolyploidy. In this latest revision we have tried to clarify even further the nature of these specific repetitive elements in the case of (tobacco, Camelina, strawberry). Specifically, we classify retrotransposons based on their protein-sequence identity compared to the consensus sequences identified by GypsyDB, as well as show that the subgenome-enriched sequences identify specific clusters of LTR-retrotransposon families that are themselves defined independently from these k-mers. Further detailed analysis of subgenome-specific transposable element families or subfamilies is beyond the scope of the present work, as it would require extensive additional work without adding any new conceptual information or clarity.

Although I spotted several small issues where arguments could be raised and incrementally revised for ever, I would like to only mention one aspect connected to my major comment above and that left me in some confusion. It was certainly a good idea to add new case studies offering pragmatic support and, despite sometimes superficial description, it mostly enabled to highlight possible caveats that the lack of theoretical background would have otherwise left hidden. More specifically, that the approach may not work properly anymore in older polyploids remains slightly unclear to me and would deserve some more detailed justification. Indeed, that TEs usually get increasingly fragmented and divergent with time is well known and logical, and this is very certainly a chief motivation for the use of short k-mers (again, something for which a justification has been requested and that is here superficially addressed). However, it remains unclear to me how the authors connect the TE life-cycle to their approach, results and conclusions. Here, the proverbial “4.5 MY half-life” of TEs is simply referenced from a single paper in grasses (although no grass was included in this study) to justify that 6-7 MY old polyploids such as Brassica do not work properly... This is at best sub-optimal, possibly misleading. Some more in depth justification would be desirable. I guess that it would be fair to say that not much is known about the rate of TE decay. Accordingly, it would be rather useful to discuss how long it may take for TE sequences to diverge to such an extent that sufficiently repeated 13-mer motifs cannot be identified anymore; for instance relying on studies along the line of Maumus & Quesneville 2014 (Ancestral repeats have shaped epigenome and genome composition for millions of years in *Arabidopsis thaliana*. *Nat Commun* 5:4104. <https://doi.org/10.1038/ncomms5104>) or simple probabilities.

Thanks for this comment. We have made every effort to address previous comments, and respond in detail below (and in revised manuscript) to this comment.

Maumus and Quesneville (2014) showed that repetitive sequences in Arabidopsis are in some cases derived from old TE families (15-30 Ma by their estimate). They follow the standard procedure of (1) clustering together repetitive elements with similar sequences into a family, (2) finding a consensus (to approximate the nominal ancestral sequence of the family), and (3) measuring sequence divergence relative to the consensus to estimate the age of the family. The fact that some TE families with relict sequences are very old, however, is not sufficient for application of our method.

Please note that our method relies on past progenitor-specific bursts of TE activity that distributed multiple copies across all chromosomes of the progenitor genome (“lightning bolts” in **Figure 1**). These copies must be (1) recognizable in the polyploid genome, and (2) differentiable at the sequence level from other, possibly similar, TEs that were active either before the progenitors diverged or after allopolyploid formation. In practice this often means we are looking at a specific “sub-family” of TE, rather than a maximally defined TE family as in Maumus and Quesneville 2014, who group together all family members that have recognizably similar sequence, without regard to chromosomal distribution.

The maximum age of sub-genome specific TE relicts that can be considered in any particular case depends not only on nucleotide substitution rates but critically on (1) the number of copies in the original burst, and (2) the half-life for deletion of a non-functional sequence. For example, in the allopolyploid frog *Xenopus laevis*, we previously identified large families of DNA transposons (Harbingers) that are specific to the two progenitors after more than 15 million years. Except for such case studies, we cannot provide first principles limits on the maximum detectable age of allopolyploids using our method, since it depends on the poorly known turnover (deletion) rate which, as you note, has only really been characterized in grasses.

We have added new heading, “Relationship of k-mers to transposable element evolution” (page 14) discussing this general issue using tobacco as an example, illustrated with **Supplementary Figure 11**. We have also provided a **Supplementary Data File 13** to further elaborate on this point using data from strawberry, Camelina, and tobacco.

Reviewers' Comments:

Reviewer #1:

Remarks to the Author:

The authors have done a thoroughly satisfactory job of addressing the final concern that I raised in my November 28, 2021 e-mail to the editors regarding proper acknowledgement of the Gordon et al., 2019 "POLYCracker" paper. On the basis of this and the authors' previous responses to my previous review comments, I am now pleased to highly recommend this fascinating and important manuscript for publication in Nature Communications.

Reviewer #5:

Remarks to the Author:

After going through the manuscript and supplementary information, I think that the concerns of Reviewer #4 had been adequately addressed. The new supplementary files support the overlap of k-mers and LTR-retrotransposons, and the extended Fig 11 exemplifies well the expected link between the k-mer approach and the progenitor-specific LTR-retrotransposon amplification. I think the evidence presented in this manuscript is convincing.